# Epigenetic control of *IL-23* expression in keratinocytes is important for chronic skin inflammation

Hui Li[1,2], Qi Yao[1,2,11], Alberto Garcia Mariscal[1,2], Xudong Wu[2,3], Justus Hülse[1,2], Esben Pedersen[1,2], Kristian Helin [2,3], Ari Waisman [4], Caroline Vinkel[5,6], Simon Francis Thomsen[5,6], Alexandra Avgustinova[7,8], Salvador Aznar Benitah [7,8], Paola Lovato[9], Hanne Norsgaard[9], Mette Sidsel Mortensen[9], Lone Veng[9], Björn Rozell[10] & Cord Brakebusch[1,2]

The chronic skin inflammation psoriasis is crucially dependent on the IL-23/IL-17 cytokine axis. Although IL-23 is expressed by psoriatic keratinocytes and immune cells, only the immune cell-derived IL-23 is believed to be disease relevant. Here we use a genetic mouse model to show that keratinocyte-produced IL-23 is sufficient to cause a chronic skin inflammation with an IL-17 profile. Furthermore, we reveal a cell-autonomous nuclear function for the actin polymerizing molecule N-WASP, which controls IL-23 expression in keratinocytes by regulating the degradation of the histone methyltransferases G9a and GLP, and H3K9 dimethylation of the IL-23 promoter. This mechanism mediates the induction of IL-23 by TNF, a known inducer of IL-23 in psoriasis. Finally, in keratinocytes of psoriatic lesions a decrease in H3K9 dimethylation correlates with increased IL-23 expression, suggesting relevance for disease. Taken together, our data describe a molecular pathway where epigenetic regulation of keratinocytes can contribute to chronic skin inflammation.

[1] Department of Biomedical Sciences, University of Copenhagen, Ole Maaløes Vej 5, 2200 Copenhagen, Denmark. [2] Biotech Research and Innovation Centre (BRIC), Ole Maaløes Vej 5, 2200 Copenhagen, Denmark. [3] Centre for Epigenetics, Ole Maaløes Vej 5, 2200 Copenhagen, Denmark. [4] Institute for Molecular Medicine, Johannes Gutenberg-University Mainz, Obere Zahlbacher Straße 67, 55131 Mainz, Germany. [5] Department of Dermatology, Copenhagen University Hospital Bispebjerg, Bispebjerg Bakke 23, 2400 Copenhagen, Denmark. [6] Department of Biomedical Sciences, University of Copenhagen, Blegdamsvej 3, 2200 Copenhagen, Denmark. [7] Institute for Research in Biomedicine (IRB Barcelona), The Barcelona Institute of Science and Technology, Barcelona 08028, Spain. [8] Catalan Institution for Research and Advanced Studies (ICREA), Barcelona 08010, Spain. [9] LEO Pharma A/S, Industriparken 55, 2750 Ballerup, Denmark. [10] Department of Experimental Medicine, University of Copenhagen, Blegdamsvej 3, 2200 Copenhagen, Denmark. [11]Present address: Center for Healthy Aging, Department of Cellular and Molecular Medicine, University of Copenhagen, Copenhagen 2200, Denmark. Correspondence and requests for materials should be addressed to C.B. (email: cord.brakebusch@bric.ku.dk)

Aberrant cytokine expression is believed to be crucial for the development of inflammatory skin diseases such as psoriasis[1]. Although the cause of psoriasis is unknown, development and maintenance of the disease occurs through the crosstalk between immune cells and keratinocytes. In particular, the interleukin (IL)-23/IL-17 axis and the tumor necrosis factor (TNF) pathway are of central importance in psoriasis as demonstrated by successful therapeutic intervention against these cytokines[2,3]. IL-23 is moreover increased in the skin of patients suffering from atopic dermatitis (AD) or alopecia areata[4] and in the serum of patients with autoimmune diseases as systemic lupus erythematosus (SLE)[5] or Crohn's disease[6].

IL-23 is required for the development and expansion of IL-17-producing immune cells, but the mechanism underlying the increased expression of IL-23 is less clear. Expression of IL-23 is regulated by nuclear factor (NF)-κB signaling, which is activated by many cytokines including TNF[7]. TNF and IL-23 are produced by activated innate immune cells, but also by psoriatic keratinocytes[8,9], which may contribute to the development of the disease. We now provide evidence that an epigenetic mechanism involving TNF and the neural Wiskott-Aldrich syndrome protein (N-WASP) controls IL-23 expression in keratinocytes.

N-WASP is a ubiquitously expressed member of the WASP/Scar family, which promotes actin polymerization via the Arp2/3 complex[10,11]. In vitro data indicate a role for N-WASP in the formation of filopodia, the assembly of intercellular junctions, and clathrin-mediated endocytosis[12–15]. In addition to these cytoplasmic functions, N-WASP might also have a nuclear role, since it was reported to shuttle between the nucleus and the cytoplasm and to bind to a nuclear complex containing actin and RNA polymerase II[16].

Genetic ablation of the N-WASP gene leads to early embryonic lethality in mice; however, filopodia formation is intact[17]. Keratinocyte-restricted deletion of the N-WASP gene revealed an important function for N-WASP in the control of the hair cycle, corresponding to increased transforming growth factor-β (TGFβ) signaling and reduction of hair follicle stem cells[18,19].

Closer investigation of the interaction between N-WASP function in keratinocytes and the immune system revealed now that loss of N-WASP in keratinocytes provokes IL-23 expression in keratinocytes and a chronic skin inflammation with an IL-17 profile. Furthermore, we observed increased amounts of auto-reactive antibodies in the serum. In vitro studies revealed that both TNF and N-WASP are regulating IL-23 expression by controlling H3K9 methylation via modulating protein degradation of the H3K9 methyltransferases G9a and GLP. These data suggest that keratinocytes might contribute to initiation or progression of inflammatory skin diseases such as psoriasis by producing IL-23 and reveal an unexpected nuclear function of N-WASP in epigenetic repression of IL-23, which is regulated by TNF.

## Results

**N-WASP ko in keratinocytes causes chronic skin inflammation.** Mice with a keratinocyte-restricted deletion of the N-WASP gene (N-WASP fl/fl K5 cre; called ko) show increased TGFβ signaling in the epidermis, suggesting the possibility of a local inflammatory reaction[18]. To investigate in more detail a potential inflammatory response in N-WASP ko mice, we performed full necropsies of 8–26-week-old control and N-WASP ko mice and analyzed tissue sections by hematoxylin–eosin (H&E) staining.

Skin sections from back, tail, eye lid ear, nose, and facial skin of ko mice of all ages indicated slight to moderate hyperplasia of the epidermis and increased cellularity of the dermis consisting of mast cells, granulocytes, and other mostly non-lymphoid cells (Fig. 1a). Staining of back skin for mast cells confirmed increased number of mast cells in N-WASP ko mice (Fig. 1g). This phenotype was particularly prominent in mechanically stressed tissues such as tail, lip, and ear, suggesting that N-WASP-null skin is oversensitive to mechanical stress. Macroscopically, the tails of older N-WASP mutant mice appeared often scaly (Fig. 1b) and mechanically stressed tissues of older mutant mice showed occasionally focal ulcerations.

Testing proliferation of interfollicular keratinocytes in N-WASP ko mice in an earlier study with only few mice, we had observed a high variation in proliferation and could not detect a significant change[18]. Repeating the proliferation measurement by testing for Ki67$^+$ cells using more mice and more samples, we now show that proliferation in N-WASP ko epidermis is indeed normal in 6- and 9-day-old mice, but increased at 3 and 7 weeks compared to littermate controls (Fig. 1d). These data indicate that N-WASP ko mice develop a skin hyperplasia after birth.

K6, which is often upregulated in hyperproliferation, was not detectable in interfollicular keratinocytes of 6-day-old N-WASP ko mice, confirming the absence of hyperplasia in very young mice. However, in 9-day-old N-WASP ko mice, rare, focal upregulation of K6 expression was observed in the interfollicular epidermis, which then became more widespread in older mice (Fig. 1c). In our earlier analysis, no interfollicular K6 was detectable in 9-day-old N-WASP ko mice[18]. The reason for this difference is not clear, but might be influenced by differences in the genetic background of the outbred mice used for the analysis and by a different animal house environment.

Next, we analyzed whether the increased cellularity of the dermis in ko mice is due to an increased number of immune cells by staining for the pan-leukocyte marker CD45 (Ly-5.2). While we could see no difference in the frequency of Ly-5.2$^+$ cells in skin of 6-day-old control and ko mice, N-WASP ko skin showed increased numbers of CD45$^+$ cells at 9 days and later indicating the presence of an leukocyte infiltrate (Fig. 1e). Fluorescence-activated cell sorting (FACS) analysis confirmed an increased number of immune cells infiltrating epidermis of N-WASP ko mice at 2 and 4 weeks (Fig. 1f).

To test whether the activation of the immune system also includes the adaptive immune response and the production of antibodies, we determined the deposition of immunoglobulin G (IgG) in the skin of control and ko mice at different ages by immunofluorescent staining and western blot. The 7-week-old N-WASP ko mice showed strongly elevated levels of IgG by western blot of skin lysates and immunofluorescent staining of back skin sections (Fig. 2a, b). In older mice, IgG deposits were also found in the kidney (Fig. 2b). No increased IgG was detectable in the skin of 3-week-old ko mice (Fig. 2a).

Next, we investigated whether inflammatory changes could also be observed in other tissues than skin. Skin-draining inguinal lymph nodes were increased in size in 8-week-old N-WASP ko mice and showed significantly increased cellularity (Fig. 2c). Furthermore, about 20% of the ko mice older than 4 months developed a mostly unilateral kidney hydronephrosis, corresponding to an inflammation of the ureter, which might lead to occlusions preventing the transport of urine from the kidney to the bladder (Fig. 2d, e). Since K5 is also expressed in ureter epithelium, K5$^-$cre-driven deletion of the N-WASP gene might also take place in ureter epithelium and mediate inflammation.

Finally, we screened serum samples (1:100) for antibodies against mouse keratinocytes. Of the 8 ko mice tested, all showed clear presence of self-reactive antibodies against nuclear (4 of 8) or cytoplasmic (4 of 8) antigens. No or clearly weaker autoreactivity was detected at this dilution in the sera of control mice (Fig. 2f). Further analysis of sera with antibodies against

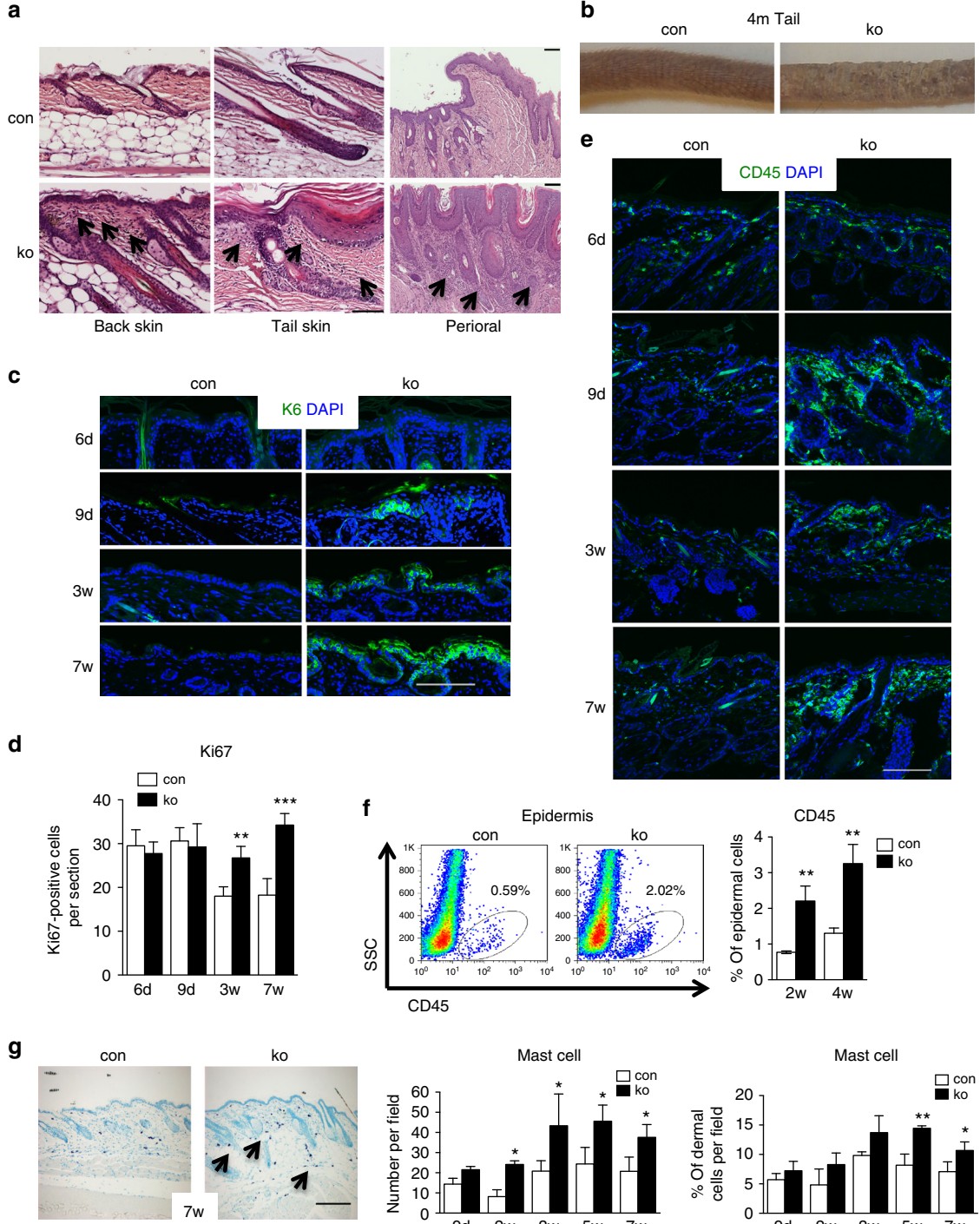

**Fig. 1** Chronic skin inflammation in mice with a keratinocyte-restricted deletion of *N-WASP*. **a** H&E-stained back, tail, and perioral skin sections of 7-week-old mice (*n*: 6/6). Arrows indicate increased dermal cellularity. **b** Representative image of tails of 4-month (4m)-old control and ko mice. **c** K6 immunofluorescence of back skin section of at indicated ages (*n*: 4/4, mean ± SD, two-tailed unpaired *t*-test). **d** Quantification of Ki67+ cells in back skin epidermis sections (*n*: 4/4, mean ± SD, two-tailed unpaired *t*-test). **e** CD45 immunofluorescence of back skin at the indicated ages (*n*: 4/4). **f** FACS for CD45+ cells in back skin epidermis and quantification (*n*: 3/3, mean ± SD, two-tailed unpaired *t*-test). **g** Representative toluidine blue staining for mast cells and quantification of mast cell infiltration in numbers per field and ratio of dermal cells per field (*n*: 3/3, two-tailed unpaired *t*-test). Arrows indicate mast cells. Scale bars (**a**, **c**, **e**, **g**): 100 μm (*$p \leq 0.05$; **$p \leq 0.01$; ***$p \leq 0.001$)

nuclear antigens revealed increased concentration of antibodies against double-stranded DNA (dsDNA) in serum samples of *N-WASP* ko mice (Fig. 2g).

These data indicate that loss of N-WASP in keratinocytes results in chronic skin inflammation together with hyperplasia of the epidermis, increased expression of K6, and increased

proliferation. Additionally, increased amounts of self-reactive antibodies are observed in serum of aged *N-WASP* ko mice.

**Increased cytokine expression in N-WASP-null back skin.** Chemokines, cytokines, and growth factors are soluble mediators involved in the activation of immune cells. We therefore analyzed

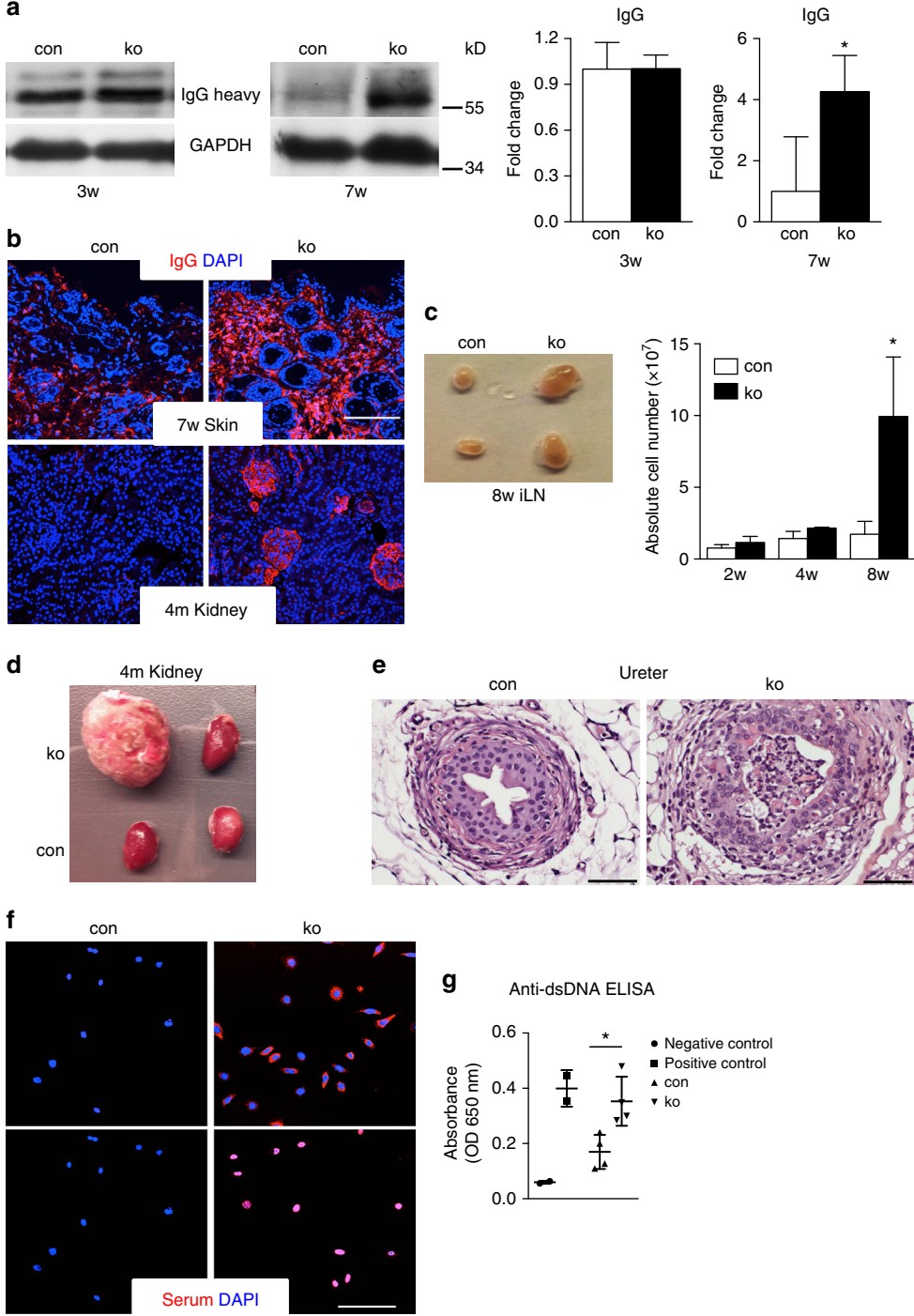

**Fig. 2** *N-WASP* knockout keratinocytes show multi-organ inflammation and autoimmunity. **a** Immunoblot analysis of IgG deposition in back skin at indicated ages. All normalized to GAPDH ($n > 3$, mean ± SD, two-tailed unpaired $t$-test). **b** Representative immunofluorescence of IgG deposition of back skin and kidney at indicated ages ($n$: 3/3, mean ± SD, two-tailed unpaired $t$-test). Scale bar: 100 μm. **c** Representative image of inguinal lymph nodes (iLN) of con and ko mouse and quantification of cellularity of iLN at indicated ages ($n$: 3/3, mean ± SD, two-tailed unpaired $t$-test). **d** Representative image of kidneys of 4-month (4m)-old control and ko mice. **e** Representative of H&E staining of ureter. Scale bar: 50 μm. **f** Representative immunofluorescence staining of serum from con and ko ($n$: 8/8). **g** Detection of anti-dsDNA antibodies in serum (1:200) by ELISA ($n$: 4/4; mean ± SD, two-tailed unpaired $t$-test; $*p \leq 0.05$)

by real-time quantitative reverse transcription-polymerase chain reaction (qRT-PCR) the expression of 17 proinflammatory mediators suggested to contribute to inflammatory skin diseases such as psoriasis, atopic dermatitis, or irritant contact dermatitis in the back skin before immune cell infiltration (6 days), at the onset of immune cell infiltration (9 days), and when IgG is

detectable in the skin (7 weeks). The 6-day-old ko mice showed increased expression of *IL-23A*, *CXCL10*, and *CCL20* (Fig. 3a) compared to controls. At 9 days, *IL-12A, IFNγ, IL-4, IL-13, IL-17A, IL-17F, IL-22, CCL2,* and *IL-1β* are also upregulated (Supplementary Figure 2a), while in 7-week-old ko mice all cytokines tested including *IL-1α, IL-6, IFNα, IFNβ,* and *TNFα* are increased

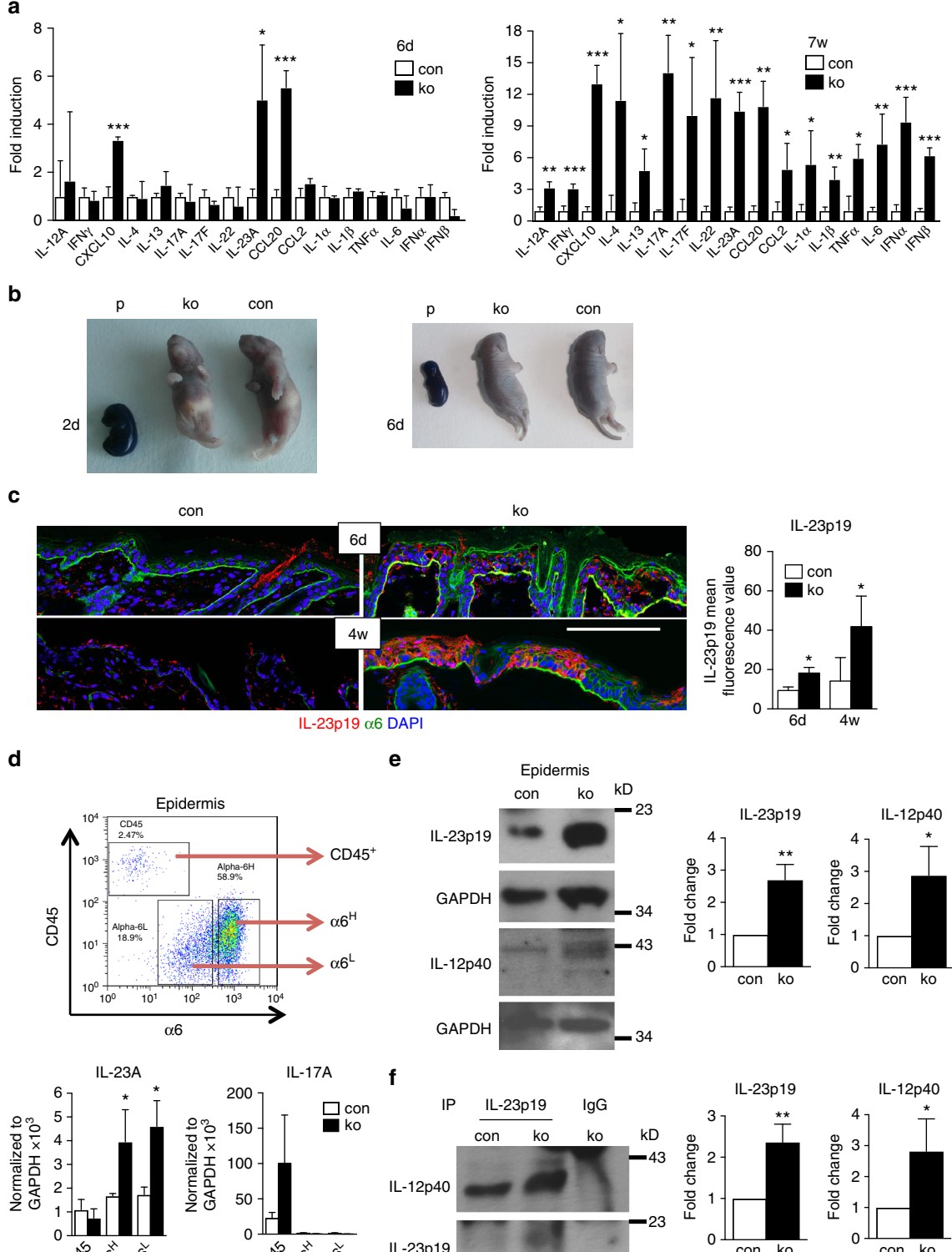

**Fig. 3** Increased cytokine expression in *N-WASP*-null skin. **a** qRT-PCR analysis of indicated genes from back skin RNA of con and ko at indicated ages (*n* > = 3/3, two-tailed unpaired *t*-test). All normalized to GAPDH. **b** Representative toluidine blue staining for barrier defect in con and ko mice at indicated ages (P: E15.5 embryos used as positive controls; 2 days (2d) *n*: 6/5, 6 days(6d) *n*: 4/5, 3 independent experiments). **c** Representative immunofluorescence staining and quantification of IL-23p19 of con and ko back skin at indicated ages (*n*: 3/3, two-tailed paired *t*-test). Scale bar: 100 μm. **d** Representative sorting strategy of CD45[+], α6[H], and α6[L] cells from con and ko epidermal cells and qRT-PCR analysis of IL-23A and IL-17A expression in sorted cells (*n*: 3/3, mean ± SD, two-tailed unpaired *t*-test, 3 independent sortings). **e** Representative immunoblots and quantification of indicated proteins from epidermal lysates of 7–8-week old con and ko mice (*n*: 4/4, mean ± SD, two-tailed unpaired *t*-test). **f** Representative immunoblots and quantification of indicated proteins of co-precipitations of IL-23p19 with IL-12p40 from epidermal lysates of 7–8-week-old con and ko mice (*n*: 4/4, mean ± SD, two-tailed unpaired *t*-test; *p ≤ 0.05; **p ≤ 0.01; ***p ≤ 0.001)

in the back skin compared to controls (Fig. 3a). IL-23 is a dimer consisting of the proteins IL-23p19 (gene: IL-23A) and IL-12p40 (gene: IL-12B). Both of these subunits are increased in *N-WASP* ko skin on the protein level (Fig. 3e), indicating increased expression of IL-23 in the absence of *N-WASP*. Co-precipitation of IL-23p19 with IL-12p40 demonstrated increased amounts for IL-23 heterodimers in *N-WASP* ko epidermis (Fig. 3f).

To investigate whether the increased cytokine expression is related to impaired barrier function and increased Toll-like receptor signaling, we performed barrier assays and tested for phosphorylation of NF-κB (pNF-κB (S536)) and pIRAK4 (T345, S346). Dye penetration assay did not indicate impaired barrier function in 2- or 6-day-old mice (Fig. 3b). Furthermore, no increased phosphorylation of NF-κB or IRAK4 could be detected in skin lysates of 6-d-old mice (Supplementary Figure 2c). These data suggest that the increased cytokine expression is not caused by a primary barrier defect. In 7-week-old *N-WASP* ko mice, expression of *involucrin*, *fillagrin* (50 ± 9.8-fold increase as determined by qRT-PCR, *n*: 2/2), *cystatin A*, *Tgm1*, and a number of small proline-rich genes and late cornified envelope genes was increased, while expression of *loricrin* and *corneodesmosin* was unchanged compared to controls, suggesting maybe subtle changes of the barrier at the stage of chronic skin inflammation (Supplementary Table 1).

Immunofluorescent staining for IL-23p19 in back skin sections of 6-day and 4-week-old control and ko mice confirmed expression of IL-23p19 in the epidermis of 6-day-old ko mice in both basal and suprabasal keratinocytes with further enhanced expression in the epidermis of 4-week-old ko mice (Fig. 3c). These data indicate that increased expression of IL-23p19 is a very early event in skin inflammation in N-WASP ko mice, which precedes the expression of most other cytokines and immune cell infiltration.

To analyze the cellular source of IL-23, epidermis of 4-week-old control and ko mice was trypsinized and the cell suspension stained and sorted by flow cytometry into immune cells (CD45[+]), basal keratinocytes (CD45[−] α6 integrin[high]), and suprabasal keratinocytes (CD45[−] α6 integrin[low]). qRT-PCR analysis revealed that the increased amounts of *IL-23A* in N-WASP-null epidermis are exclusively produced by basal and suprabasal keratinocytes and not by immune cells (Fig. 3d). As expected, *IL-17A* is only expressed by immune cells, but neither by control nor *N-WASP*-null keratinocytes (Fig. 3d).

Detection of IL-12, IL-1β, mKC, and IL-17F protein at different time points by enzyme-linked immunosorbent assay (ELISA) analysis confirmed a time-dependent increase in cytokine expression in skin, tail, and ear of *N-WASP* ko mice (Supplementary Figure 1). In addition to these data we have previously reported that TGFβ-dependent Smad2 phosphorylation is increased in epidermis of 3-week-old N-WASP ko mice[18], suggesting that TGFβ might also contribute to the development of skin inflammation. However, Smad2 phosphorylation was not increased in 9-day-old ko mice (Supplementary Figure 2d).

These data show that *N-WASP*-null keratinocytes produce innate cytokines and chemokines including IL-23 before infiltration of immune cells. At around 9 days of age, infiltrating immune cells contribute to the production of a range of cytokines and chemokines, reflecting a broad immune activation. Later, IL-17 expression increases establishing the IL-23/IL-17 axis.

**Different immune cells contribute to IL-17 in *N-WASP* ko mice**. Since IL-17A-producing cells are of critical importance in psoriasis and probably also in other chronic inflammatory diseases such as AD and SLE[20], we wanted to confirm the production of IL-17A on the protein level and identify the cells

producing this cytokine. For this reason we tested epidermal cell suspensions of 4-week-old mice by FACS for IL-17A-producing leukocytes and investigated skin-draining inguinal lymph nodes of 2- and 4-week-old control and ko mice for intracellular IL-17A and the surface markers CD45, CD3, CD4, and γδTCR. In the epidermis of 4-week-old ko mice, only a small percentage of the leukocytes produced IL-17A, about 4 times more than in controls (Fig. 4a). In skin-draining inguinal lymph nodes of 2- and 4-week-old ko mice, frequency and absolute numbers of CD4 cells and γδ T cells expressing IL-17A was strongly increased (Fig. 4b, c). No significant increase was observed in mesenteric lymph nodes of 4-week-old mice (Fig. 4g), suggesting that the IL-17A production is a consequence of the skin inflammation and not due to a systemic defect in T-cell development. These data show that different T-cell subsets contribute to the increased IL-17 levels in *N-WASP* ko mice.

Since K5 promoter-driven Cre expression might also lead to deletion of the N-WASP gene in thymic epithelium which is important for thymocyte selection, we determined the sizes of thymocyte subpopulations (DN, DP, CD4SP, CD8SP) and of CD4 and CD8 T cells in spleen and bone marrow in 9-day-old *N-WASP* ko and control mice. Furthermore, thymus epithelium was histologically analyzed by H&E staining. No difference was found between control and *N-WASP* ko mice in these assays (Fig. 4d–f), making it very unlikely that altered thymocyte development is contributing to the skin inflammation phenotype.

**Skin inflammation is dependent on keratinocyte-derived IL-23**. To test whether the skin inflammation observed in *N-WASP* ko mice is dependent on keratinocyte-derived IL-23, we generated mice with a conditional knockout of *IL-23A* using CRISPR/Cas9 genome editing (Fig. 5a), and crossed them with N-WASP ko mice. The resulting mice with a keratinocyte-restricted loss of both *N-WASP* and *IL-23A* (dko) showed a dramatically reduced expression of *IL-23A* messenger RNA (mRNA), indicating efficient deletion of the *IL-23A* gene in keratinocytes (Fig. 5d). Importantly, infiltration of leukocytes, keratinocyte proliferation (Ki67[+] cells), and K6 expression were strongly reduced compared to *N-WASP* ko mice (Fig. 5b, c). Interestingly, dko mice still showed epidermal thickening and loss of hairs. The reduced inflammation in dko mice correlated with a decreased expression of *IL-17A* and *S100A9* in skin (Fig. 5d). Average values of *IL-17F, IL-22, S100A8, CCL2, CCL20, CXCL10*, and *IFNγ* were not altered in dko compared to ko, while *IL-12A* and *IL-12B* were strongly reduced. *LL37*, which was described to be a key factor in activating plasmacytoid dendritic cells in psoriasis[21], *IL-19, IL-20*, and *IL-24* were even increased in dko mice.

Repeating the gene expression analysis with FACS purified keratinocytes, we obtained similar results as for total skin for *S100A8, S100A9, CCL2, CCL20, CXCL10, IL-12B*, and *IL-23A* (Fig. 6a). Interestingly, average expression of *IL-17F, IL-19*, and *LL37* was decreased in both ko and dko keratinocytes compared to controls, although the changes were not significant. Low expression of these mediators indicated that they are mainly produced by immune cells. In keratinocytes, no expression was detected for *IL-17A, IL-22*, and *IFNγ*.

These data demonstrate that the skin inflammation in N-WASP ko mice is dependent on keratinocyte-produced IL-23.

**Primary *N-WASP*-null keratinocytes express IL-23A in vitro**. To understand the molecular mechanism underlying the increased *IL-23A* expression in *N-WASP*-null keratinocytes in vivo, we tested whether *IL-23A* expression is also increased in cultured keratinocytes lacking N-WASP. We therefore isolated primary keratinocytes from 3-day-old control and ko mice,

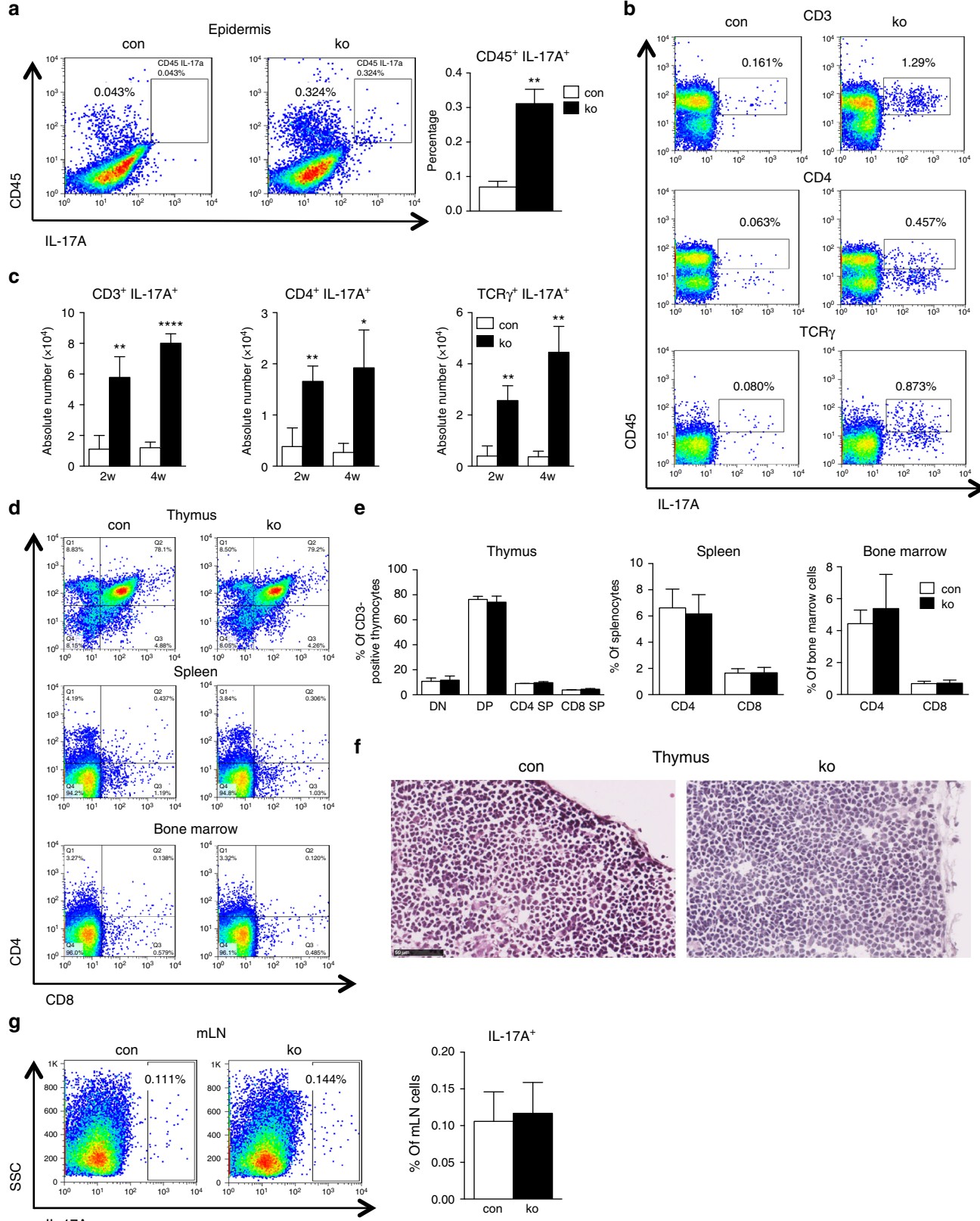

**Fig. 4** Increased IL-17A-producing cells in *N-WASP* ko mice. **a** Representative and quantification of intracellular FACS analysis of IL-17A[+] cells in epidermis (*n*: 3/3, mean ± SD, two-tailed unpaired *t*-test). **b**, **c** Representative and quantification of intracellular FACS analysis of IL-17A[+] cells in iLN (*n*: 3/3, mean ± SD, two-tailed unpaired *t*-test). **d**, **e** Representative and quantification of CD4, CD8 distribution in thymus, spleen, and bone marrow from 9-day-old con and ko mice (*n*: 3/3, two-tailed unpaired *t*-test). **f** Representative H&E staining of thymus from 9-day-old con and ko mice (*n*: 3/3). Scale bar: 50 μm. **g** Representative and quantification of intracellular FACS analysis of IL-17A[+] cells in mLN (*n*: 6/6, mean ± SD, two-tailed unpaired *t*-test; *\*p* ≤ = 0.05; *\*\*p* ≤ = 0.01; *\*\*\*\*p* ≤ = 0.0001)

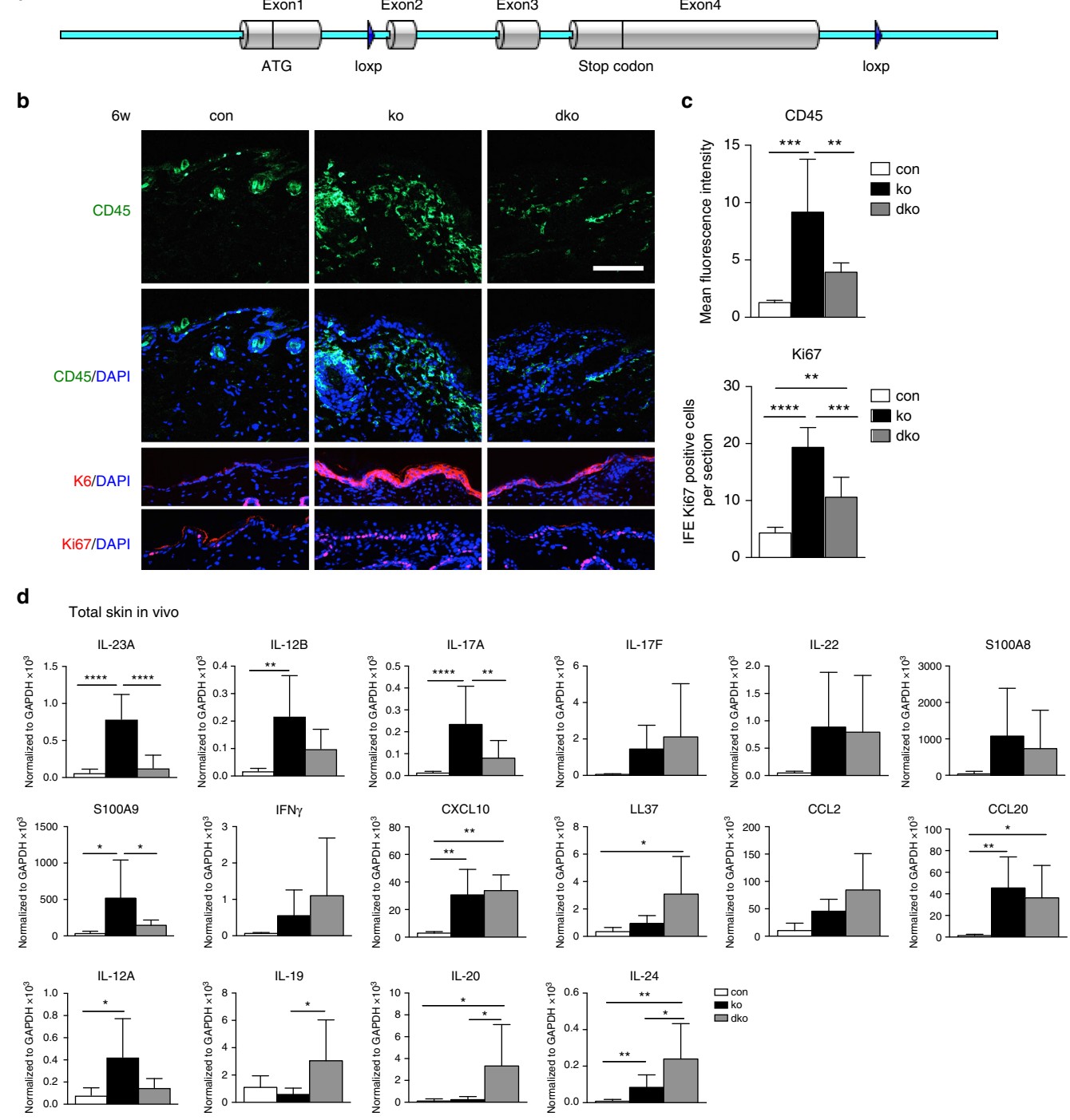

**Fig. 5** Keratinocyte-derived IL-23p19 is important for skin inflammation in *N-WASP* knockout. **a** Strategy of loxP insertion in mouse IL-23A gene. **b**, **c** CD45, K6, Ki67 immunofluorescence of back skin section from 6-week-old con, ko, and dko (**b**), quantification of mean fluorescence intensity of CD45, and quantification of Ki67-positive cells in interfollicular epidermis (IFE) (*n*: 5/5/5, mean ± SD, one-way ANOVA, with Tukey's multiple comparisons). Scale bar: 100 μm. **d** qRT-PCR analysis of indicated genes from back skin RNA 6–8-week-old of con, ko, and dko mice. All normalized to GAPDH (*n* > = 6/6/6, mean ± SD, one-way ANOVA, with Tukey's multiple comparisons; *$p \leq 0.05$; **$p \leq 0.01$; ***$p \leq 0.001$; ****$p \leq 0.0001$)

cultured them for 5–7 days in vitro and then analyzed by qRT-PCR the expression of various cytokines. We observed increased expression of *IL-23A*, *CXCL10*, and *CCL2*, while *CCL20*, *TNF*, *IL-1α*, *IL-1β*, *IL-6*, *S100A8*, and *CXCL5* were not increased (Fig. 6b). Culturing primary keratinocytes under differentiating high-calcium conditions did not change *IL-23A* expression in control or *N-WASP* ko cells (Fig. 6c).

These data suggest that *N-WASP* also controls expression of *IL-23A* in cultured neonatal keratinocytes. Analysis of gene expression by microarray analysis from primary keratinocytes isolated from 3-week-old mice[18] did not reveal any additional cytokine genes that showed a more than 1.5-fold alteration of the average value, supporting that increased *IL-23A* expression by

keratinocytes is crucial for triggering the complex inflammation observed in adult *N-WASP* ko mice.

The establishment of an in vitro system allowed us therefore to study the molecular mechanism underlying the increased *IL-23A* expression *N-WASP*-null keratinocytes in a less complex model system.

**N-WASP regulates IL-23 expression by H3K9 dimethylation.** N-WASP is known to promote actin polymerization, which is also crucial for the N-WASP-dependent regulation of RNA polymerase II-mediated transcription[16,22] and retinoic acid-induced expression of *HoxB*[23]. For this reason we tested whether treatment of keratinocytes with the actin polymerization

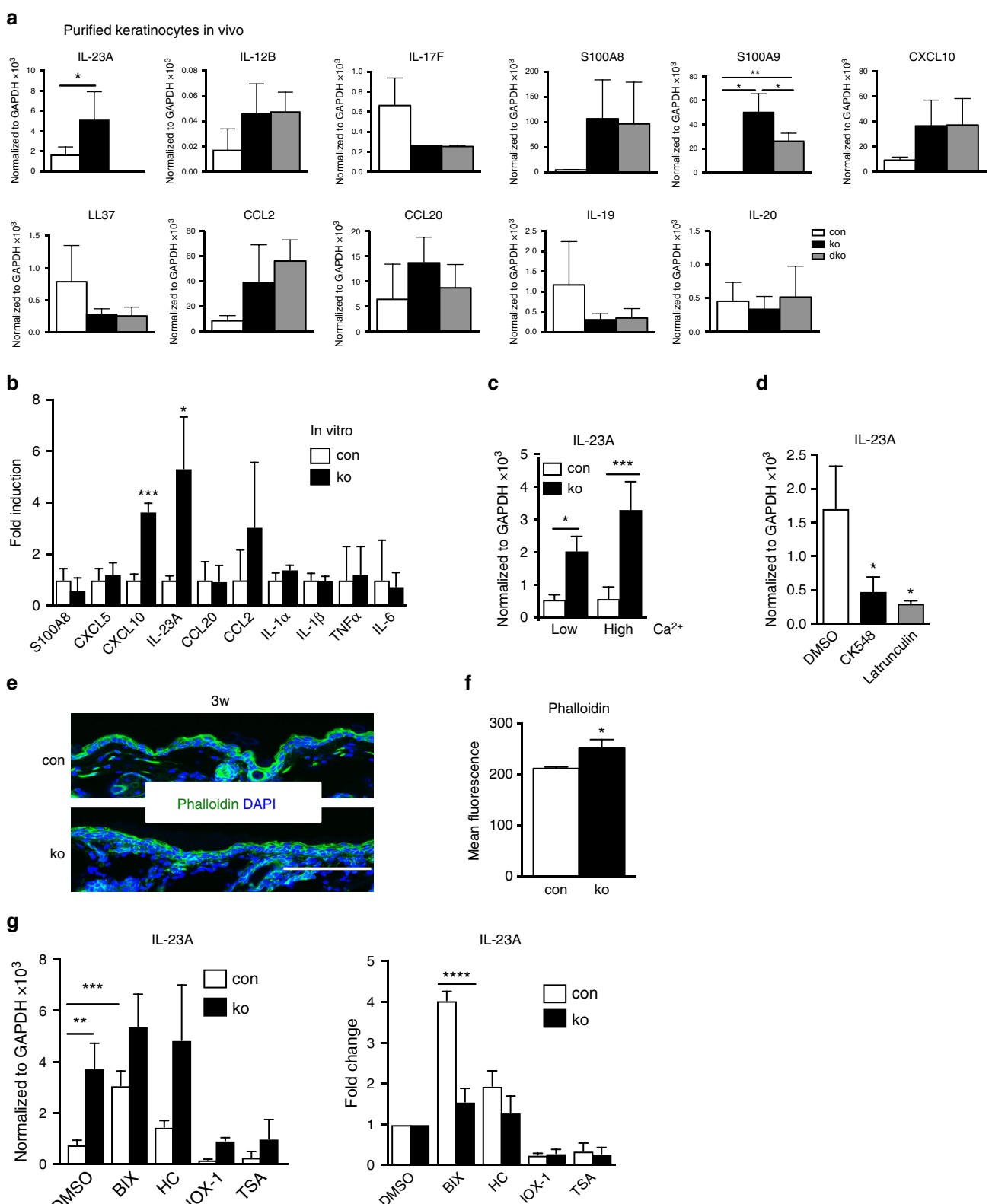

inhibitor latrunculin or the Arp2/3 inhibitor CK548 increases *IL-23A* expression in these cells. However, both inhibitors strongly reduced *IL-23A* expression (Fig. 6d). In addition, we determined the amount of F-actin in primary control and N-WASP-null keratinocytes by fluorescence microscopy and FACS analysis after staining with phalloidin–fluorescein isothiocyanate (FITC). Absence of N-WASP did not reduce the total amount of F-actin in keratinocytes (Fig. 6e, f). Taken together, these data suggest that loss of N-WASP is not promoting IL-23A expression by decreased F-actin.

The N-WASP-related molecule WASP, which is only expressed in hematopoietic cells, is interacting with epigenetic enzymes leading to increased activity of RBBP5-dependent H3K4 methyltransferases and the JMJD2A demethylase and elevated gene expression[24]. To test whether N-WASP regulates *IL-23A* expression by altering epigenetic histone marks, we treated primary keratinocytes with different epigenetic inhibitors: BIX01294 (BIX), inhibiting the H3K9 dimethyltransferases G9a and GLP and thus decreasing the repressive H3K9me2, HC (tranylcypromine hydrochloride), which blocks the lysine demethylase LSD1 and increases particularly H3K4 methylation, IOX-1, which is an inhibitor of several JmjC-domain containing histone demethylases, and trichostatin A, a histone deacetylase inhibitor, which increases histone acetylation.

The H3K9 methyltransferase inhibitor BIX induced *IL-23A* expression in control keratinocytes, while it had only a relatively small effect on N-WASP-deficient cells (Fig. 6g; right panel shows fold change relative to DMSO). One explanation for this low effect of BIX on *N-WASP*-null cells could be that H3K9 methylation is already decreased in untreated N-WASP-deficient keratinocytes. Indeed, western blot analysis revealed that *N-WASP* ko keratinocytes display 50% reduced H3K9me2 levels than control cells (Fig. 7a) Treatment with BIX reduced H3K9me2 levels in controls by about 2-fold, while it had no significant influence on ko cells.

A crucial role for H3K9me2 in the regulation of *IL-23A* expression in keratinocytes was supported by the result that IOX-1 treatment, which increases H3K9me2 (Fig. 7b), strongly reduced *IL-23A* expression in both control and *N-WASP* ko keratinocytes (Fig. 6g). Interestingly, IOX-1-treated *N-WASP* ko cells showed H3K9me2 and *IL-23A* levels similar to untreated control cells.

Also, in vivo, H3K9me2 was reduced in *N-WASP* ko keratinocytes as shown by western blot analysis of epidermal lysates of 6-day-old control and mutant mice and by immunofluorescent staining (Fig. 7c, d).

To interfere with H3K9 methylation by a different method and to test the role of this modification for the regulation of *IL-23A* expression in vivo, we analyzed IL-23 p19 expression in FACS sorted, interfollicular epidermal keratinocyte stem cells of mice with a keratinocyte-restricted deletion of the G9a gene. G9a ko keratinocytes showed increased expression of *IL-23A* to a similar extent in *N-WASP* ko cells (Fig. 7e).

To assess whether reduced H3K9 dimethylation is directly affecting the *IL-23A* promoter or only indirectly controlling IL-23A expression by regulating expression of transcription factors, a chromatin Immunoprecipitation (ChIP) analysis for different regions of the *IL-23A* promoter was performed. Indeed, we could identify an 0.3 kb region about 0.5 kb upstream of the transcription start site of *IL-23A* which showed significantly reduced H3K9me2 in the absence of N-WASP (Fig. 7f, "B"). This promoter region contains binding sites for IRF3, Smad3, and ATF-2, which were found previously to be critical for *IL-23A* expression[25,26]. Other IL-23A promoter regions tested showed no detectable alteration in H3K9 dimethylation (Fig. 4j; "A", "C").

Deleting the *N-WASP* gene in vitro by Cre expression in fl/fl primary keratinocytes decreased protein amounts of G9a and H3K9me2 and increased mRNA expression of *IL-23A*, confirming that this mechanism is cell autonomous (Fig. 7g, h).

These findings indicate that N-WASP, H3K9me2, and the H3K9 methyltransferases G9a and GLP are crucially regulating *IL-23A* expression in keratinocytes in vitro and in vivo.

**N-WASP regulates protein stability of G9a and GLP.** Since loss of N-WASP had a similar effect on *IL-23A* expression as inhibition of the H3K9 dimethylation transferases G9a and GLP, N-WASP might regulate expression, degradation, or functional activity of G9a and GLP. Gene expression analysis of *N-WASP*-null epidermis and N-WASP-null keratinocytes by microarray, however, did not indicate decreased mRNA level of G9a and GLP. qRT-PCR of epidermal and primary keratinocyte RNA for *G9a* and *GLP* confirmed that the mRNA levels are not significantly altered in vivo and in vitro (Fig. 8a, b). We therefore checked whether loss of N-WASP decreases G9a and GLP protein.

Western blots for G9a and GLP revealed decreased protein levels in lysates of cultured N-WASP-null keratinocytes and epidermal lysates of 6-day-old mice, suggesting that N-WASP is involved in regulating the stability of G9a and GLP (Fig. 8c, d). This was confirmed by analysis of the protein degradation kinetic in the presence of cycloheximide, which revealed an accelerated degradation of G9a and GLP in the absence of N-WASP (Fig. 8e).

**N-WASP is associated with chromatin-bound G9a and GLP.** Based on these results we tried to rescue the H3K9 methylation defect by treating N-WASP-null keratinocytes with the proteasome inhibitor MG-132. In addition, we assessed whether loss of N-WASP affects chromatin binding of G9a and GLP, hypothesizing that methyltransferases not attached to chromatin have a reduced half-life. To this end, keratinocytes were fractionated into a soluble cytoplasmic (C; marker: glyceraldehyde 3-phosphate dehydrogenase (GAPDH)), a soluble nuclear (SN; marker: proliferating cell nuclear antigen (PCNA)), and a pellet fraction containing chromatin (P; marker: H3). N-WASP protein was found in the cytoplasmic ($65\% \pm 4\%$; *n*: 3) and in the chromatin

**Fig. 6** *IL-23A* expression is increased in *N-WASP*-null keratinocytes. **a** qRT-PCR analysis of indicated genes in FACS purified primary keratinocytes from 6-week-old mice (*n*: 4/4/4, mean ± SD, two-tailed unpaired *t*-test, 2 independent sorting). **b** qRT-PCR analysis of indicated cytokine mRNA expression in primary keratinocytes. All normalized to GAPDH (*n*: 3/3, mean ± SD, two-tailed unpaired *t*-test). **c** Primary keratinocytes were differentiated with 2 mM CaCl$_2$ (High Ca$^{2+}$) for 48 h and then analyzed by qRT-PCR for IL-23A expression, compared to untreated cells (Low Ca$^{2+}$; *n*: 3/3, mean ± SD, two-tailed unpaired *t*-test). **d** qRT-PCR analysis of IL-23A mRNA expression in primary keratinocytes in response to 24 h treatment with indicated inhibitors. All normalized to GAPDH (*n*: 3/3, mean ± SD, two-tailed unpaired *t*-test). **e** Representative immunofluorescence of phalloidin in con and ko back skin. Scale bar: 100 μm. **f** Quantification of FACS mean fluorescence intensity of phalloidin in con and ko primary keratinocyte (*n*: 3/3, mean ± SD, two-tailed unpaired *t*-test). **g** qRT-PCR analysis of IL-23A mRNA expression in primary keratinocytes treated with indicated inhibitors for 24 h. All normalized to GAPDH. The right panel shows the same data, but normalized to the respective DMSO-treated samples, thus showing the relative effect of the treatment as fold change (*n*: 3/3, mean ± SD, two-tailed unpaired *t*-test; *$p \leq 0.05$; **$p \leq 0.01$; ***$p \leq 0.001$; ****$p \leq 0.0001$)

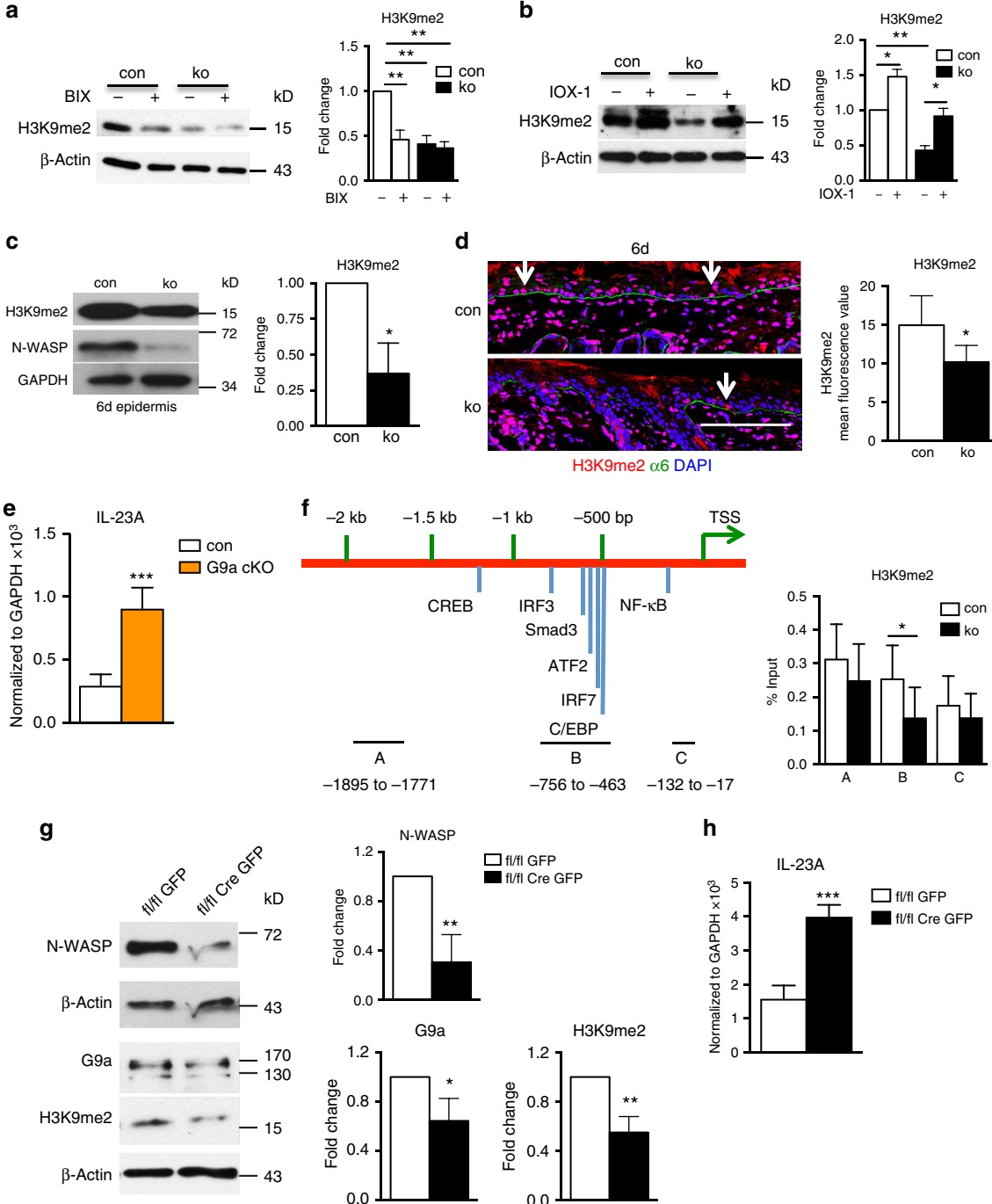

**Fig. 7** N-WASP represses IL-23p19 expression in keratinocytes by regulating H3K9 dimethylation. **a**, **b** Immunoblot analysis of H3K9me2 level in con and ko primary keratinocytes treated with BIX01294 (**a**) or IOX-1 (**b**) for 24 h (n: 3/3, mean ± SD, one-way ANOVA, with Tukey's multiple comparisons). **c** Immunoblot for H3K9me2 in epidermis of 6-day-old mice (n: 3/3, mean ± SD, two-tailed unpaired t-test). **d** H3K9me2 IF of back skin of 6-day-old mice (n: 3/3). Scale bar: 100 μm. **e** qRT-PCR analysis of IL-23A mRNA expression in sorted interfollicular epidermal keratinocytes from con and G9a knockout. All normalized to GAPDH (n: 4/4, mean ± SD, two-tailed unpaired t-test). **f** ChIP of primary keratinocytes for H3K9me2 with qPCR for indicated regions of IL-23 promoter (n: 7/7, mean ± SD, two-tailed unpaired t-test). **g**, **h** Primary *N-WASP* fl/fl cells lentivirally transduced with GFP or Cre-GFP were analyzed by western blot for indicated genes (**g**) and by qRT-PCR for IL-23A expression (**h**; n: 4/4, mean ± SD, two-tailed unpaired t-test; *p ≤ 0.05; **p ≤ 0.01; ***p ≤ 0.001)

fraction (33% ± 2%), while the soluble nuclear fraction contained only low amounts (2% ± 0.5%) (Fig. 8f).

Treatment of control and ko cells with MG-132 increased the apparent molecular weight of G9a and GLP, which might reflect ubiquitination of the proteins (Fig. 8f). In addition, protein amounts of G9a and GLP in N-WASP ko keratinocytes treated

with MG-132 increased, corresponding to normal levels of H3K9me2. Moreover, MG-132 also increased the protein level of N-WASP in the soluble nuclear and the chromatin fraction, but not in the cytosol, indicating that nuclear N-WASP is more prone to proteasomal degradation (Fig. 8f). G9a and GLP were nearly exclusively found in the chromatin pellet fraction and loss

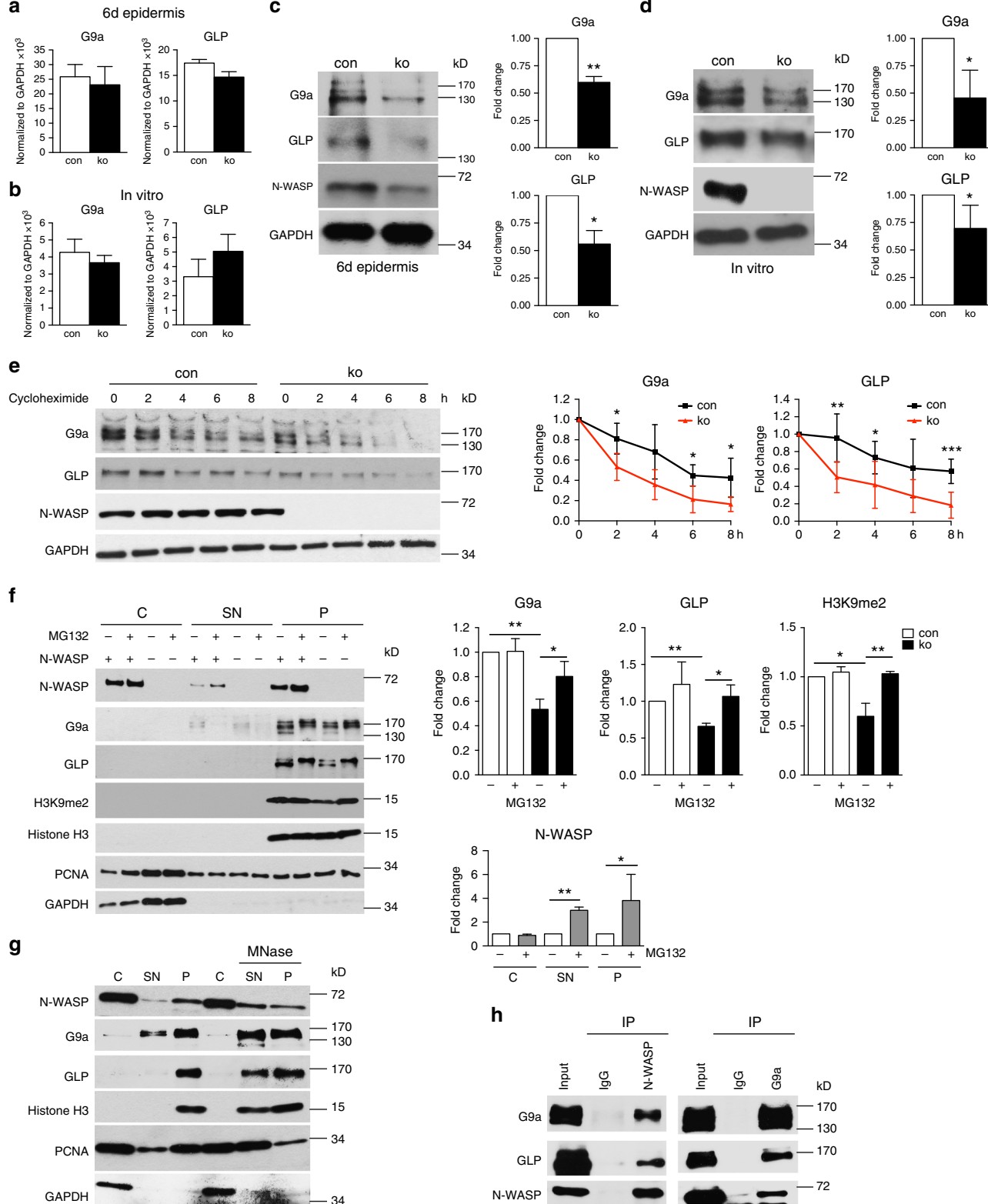

**Fig. 8** N-WASP co-precipitates with and regulates the stability of the H3K9 methyltransferases G9a and GLP. Expression of G9a and GLP mRNA (**a**, **b**) and protein (**c**, **d**) in epidermis of 6-day-old mice (**a**, **c**) and in primary keratinocytes (**b**, **d**). (*n* (**a**, **b**, **c**): 3/3; *n* (**d**): 5/5). **e** immunoblot analysis of protein stability of G9a and GLP in primary keratinocytes in the presence of cycloheximide (G9a *n*: 4/4, GLP *n*: 6/6, mean ± SD, two-tailed unpaired *t*-test). **f** Cell fractionation of keratinocytes after 90 min of treatment with the proteasome inhibitor MG-132 followed by immunoblot for indicated proteins, the indicated protein levels were quantified on the right (*n*: 3/3, mean ± SD, two-tailed unpaired *t*-test). **g** Cell fractionation of primary keratinocytes with treatment of the pellet fraction with micrococcal nuclease followed by immunoblot for indicated proteins (MNase) (*n*: 3). **h** Co-precipitation of N-WASP with G9a and GLP in primary keratinocyte nuclear lysates (*n*: 3). C: cytosolic fraction, SN: soluble nuclear fraction, P: pallet (*$p \leq 0.05$; **$p \leq 0.01$; ***$p \leq 0.001$)

of N-WASP did not increase G9a and GLP levels detected in the soluble nuclear fraction, indicating that N-WASP is not required for the chromatin association of the methyltransferases.

To confirm that N-WASP is indeed bound to chromatin and not to insoluble cytoskeleton, we repeated the fractionation, but digested chromatin with micrococcal nuclease (Fig. 8g). This treatment increased the amount of N-WASP, G9a, GLP, H3, and PCNA in the soluble nuclear fraction, indicating that N-WASP is strongly associated with chromatin, similar to G9a and GLP.

We then tested whether N-WASP is forming a complex with G9a and GLP. Indeed, N-WASP co-immunoprecipitated with both H3K9 dimethyltransferases (Fig. 8h). Reciprocally, G9a immunoprecipitated together with GLP and N-WASP. Control precipitation with IgG did not result in detectable amounts of G9a, GLP, or N-WASP. These data suggest the possibility that direct or indirect association of N-WASP with chromatin-bound G9a-GLP heterodimers is preventing the degradation of the methyltransferases.

**TNF regulates IL-23 expression via H3K9 methylation**. In psoriasis, TNF is an important activator of IL-23 expression[27]. We therefore speculated that TNF is able to induce *IL-23A* expression in keratinocytes and that this is mediated by N-WASP-dependent changes in H3K9 dimethylation (H3K9me2).

To test this hypothesis, we stimulated both control and N-WASP-null keratinocytes with TNF and checked IL-23A expression after 4 h by qRT-PCR. While TNF treatment of control cells increased *IL-23A* expression to a level comparable to N-WASP-deficient keratinocytes, it only slightly increased expression in N-WASP-null cells (Fig. 9a). Normalizing the values for TNF-treated cells by the respective values for untreated cells showed that *N-WASP*-null keratinocytes are hardly responding to TNF with respect to *IL-23A* expression, suggesting that TNF and N-WASP regulate *IL-23A* expression by the same pathway.

To examine whether this effect is caused by an increased TNF signaling in *N-WASP*-null cells, we tested surface levels of TNF-RI and activation of several TNF effectors in *N-WASP*-null and control keratinocytes. However, FACS analysis did not reveal a significant change of TNF-RI at the cell membrane (Fig. 9b). Furthermore, TNF induced phosphorylation of the TNF effector molecules p38, c-Jun N-terminal kinase (JNK), and NF-κB was unchanged in *N-WASP* ko cells compared to control keratinocytes (Fig. 9c).

We then checked how treatment with TNF is affecting protein levels of G9a and GLP and H3K9 dimethylation in control and *N-WASP*-null keratinocytes. In control keratinocytes, TNF resulted in a dose-dependent decrease of G9a, GLP, and H3K9me2 levels already after 10 min (Fig. 9c). H3K9me2 slowly increased thereafter, while G9a and GLP remained also after 4 h at a low level. *N-WASP*-null keratinocytes displayed low amounts of G9a, GLP, and H3K9 dimethylation, which did not decrease further after treatment with TNF, further supporting that TNF and N-WASP work along the same pathway (Fig. 9c).

**TNF induces phosphorylation of N-WASP**. If N-WASP is acting downstream of TNF, how is TNF signaling modifying nuclear N-WASP function? TNF activates different serine/threonine kinases as for example PKCα or IKKα. N-WASP, on the other hand, was shown to be phosphorylated by casein kinase II at serine 480/481, which decreases N-WASP-dependent actin polymerization[14]. Therefore, we speculated that TNF-induced serine kinase activation might lead to phosphorylation of N-WASP at serine 480/481 affecting the nuclear function of N-WASP. To investigate this, we fractionated keratinocytes 10 min after treatment with TNF. The chromatin fraction was extracted with a 300 mM NaCl

buffer to elute weakly chromatin-associated proteins. Treatment with TNF resulted in a shift of nuclear N-WASP from the strongly chromatin-bound fraction (Fig. 9d; P300) to the weakly chromatin-bound fraction (S300). Interestingly, within the time frame of the experiment, no significant increase of N-WASP was observed in the cytoplasmic fraction (C). Clearly, TNF stimulation reduced chromatin association of N-WASP.

Moreover, TNF increased phosphorylation of N-WASP at serine 480/481 (Fig. 6d). Phosphorylated N-WASP was found in the cytoplasmic (C) and the weakly associated chromatin fraction (S300), but not in the strongly associated nuclear fraction (P300). These data suggest that phosphorylation of N-WASP at serine 480/481 is incompatible with strong chromatin binding. Since our previous experiments demonstrated an increased proteolytic degradation of G9a-GLP heterodimers in the absence of N-WASP, the decreased chromatin attachment of phosphorylated N-WASP might be associated with a dissociation of N-WASP from the G9a-GLP heterodimers, resulting in increased degradation of the H3K9 methyltransferases.

Indeed, in a ChIP experiment, TNF treatment for 10 min resulted in a decreased H3K9me2 of the IL-23A promoter in the same region (B) as loss of N-WASP decreased H3K9me2 (Fig. 9e).

IOX-1 treatment of N-WASP ko keratinocytes rescued H3K9me2 and *IL-23A* expression (Figs. 6g and 7b), but it did not restore the ability of TNF to increase *IL-23A* expression (Fig. 9f). This inability corresponds to the unchanged levels of G9a, GLP, and H3K9me2 in TNF-treated *N-WASP* ko keratinocytes, suggesting that the N-WASP/G9a/GLP/H3K9me2 pathway is crucial for the stimulation of IL-23A expression by TNF.

Collectively, these data indicate that regulation of H3K9me2 by N-WASP is crucial for TNF-induced *IL-23A* expression in keratinocytes in mice.

**H3K9me2 is regulating IL-23 expression in human keratinocytes**. Since our previous data indicated an important role for H3K9 methylation for the regulation of *IL-23A* expression in mouse keratinocytes, we wanted to test whether this mechanism is also relevant in human keratinocytes and in psoriasis, an IL-23-dependent chronic skin inflammation.

Treatment of primary human keratinocytes for 30 min with TNF significantly decreased G9a and H3K9me2 levels, while it increased NF-κB phosphorylation (Supplementary Figure 3e). Treatment with the G9a/GLP inhibitor BIX for 24 h decreased H3K9me2 levels and increased *IL-23A* expression in a concentration-dependent manner (Supplementary Figure 3a). On the other hand, H3K9me2 levels were increased and *IL-23A* mRNA levels decreased by incubation for 24 h with the histone-demethylase inhibitor IOX-1 (Supplementary Figure 3c). Importantly, TNF-induced *IL-23A* expression was completely blocked by pretreatment for 24 h with IOX-1, while BIX pretreatment increased TNF-induced *IL-23A* expression (Supplementary Figure 3b, d).

These data demonstrate that in human keratinocytes H3K9me2 levels are also crucially important for regulating basal and TNF-induced IL-23A expression.

Furthermore, we stained three sections of lesional and non-lesional skin from three psoriasis patients for H3K9me2. Compared to skin from healthy donors, skin from psoriasis patients showed increased proliferation (Ki67; Fig. 10a). While non-lesional epidermis showed a relatively uniform nuclear staining of H3K9me2, H3K9me2 was decreased by about 50% in psoriatic lesions, particularly in the suprabasal region (Fig. 10a, c). Interestingly, skin of healthy controls stained in parallel presented two-fold higher amounts of H3K9me2 than non-lesional skin of psoriasis patients. Co-staining for IL-23p19 revealed an inverse correlation of H3K9me2 and IL-23 as

expression of IL-23p19 was increased in H3K9me2 low suprabasal regions of psoriatic lesions (Fig. 10b, c; Supplementary Figure 2b).

These correlative findings support a role of H3K9me2 in the regulation of IL-23 expression in keratinocytes in psoriasis.

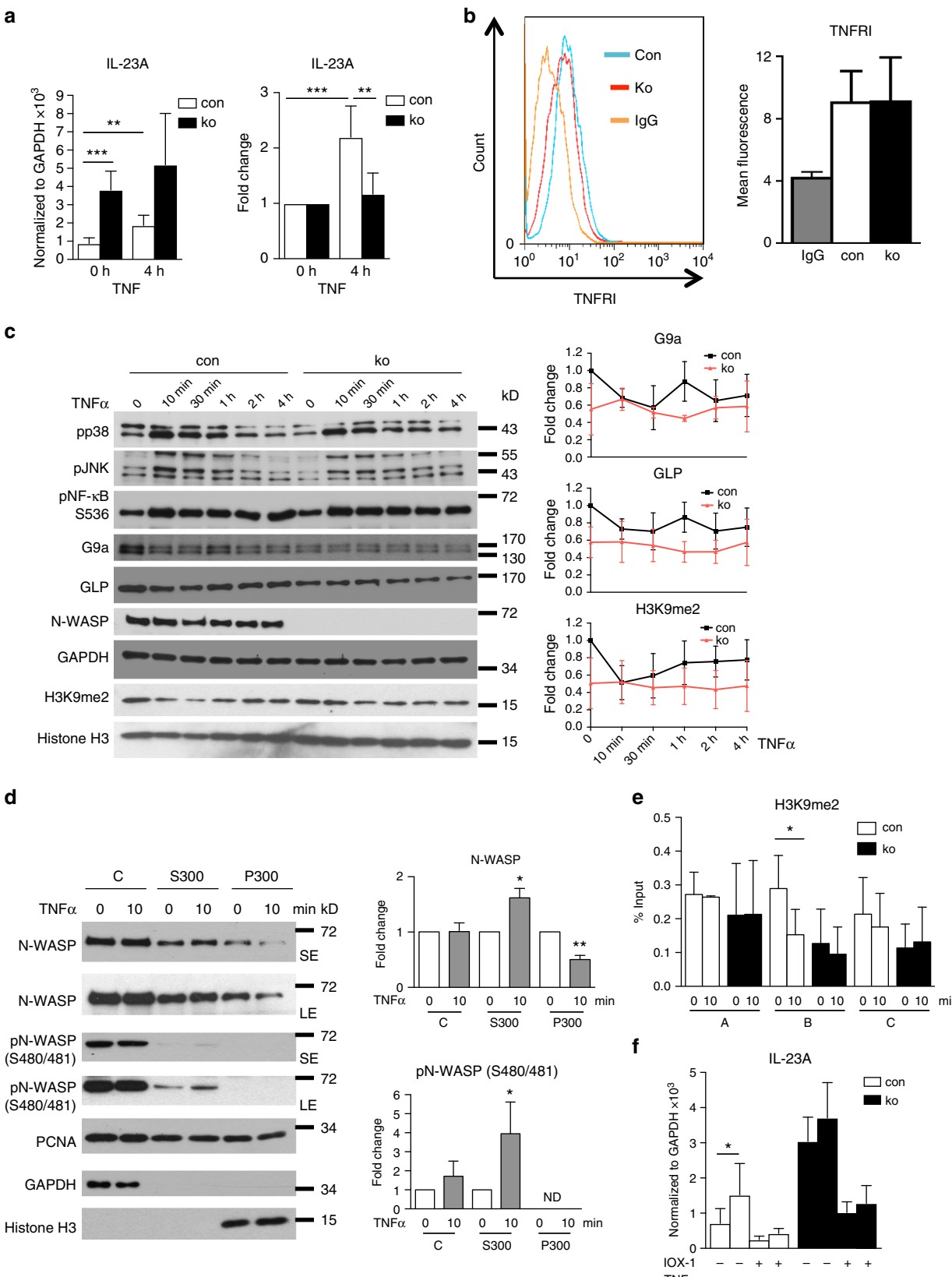

## Discussion

The TNF/IL-23/IL-17 cytokine cascade is suggested to be important for different inflammatory diseases. In psoriasis, inhibitors against these cytokines are already used in the therapy of this chronic skin inflammation[2,3]. Here we show in a mouse model that keratinocytes can produce IL-23 at levels sufficient to cause differentiation of IL-17A-producing T cells and skin inflammation. We furthermore describe a TNF/N-WASP/G9a, GLP/H3K9me2 pathway that crucially regulates IL-23 expression in human and mouse keratinocytes (Supplementary Figure 3F). Since DNA damage response leads to decreased G9a levels[28], and polyinosinic:polycytidylic acid-induced Toll-like receptor signaling decreases H3K9 methylation and increases IL-23 expression in dendritic cells[29], other signaling pathways might also converge on this epigenetic control of IL-23 production in keratinocytes.

As IL-23 is conceived to be also important in other inflammatory diseases such as AD[30] or SLE[31], epigenetic regulation of IL-23 expression in non-immune cells could also play a role in these illnesses. Indeed, antibodies against dsDNA observed in the serum of N-WASP ko mice are well known in SLE, but not in psoriasis.

In conclusion, we report here a nuclear mechanism for TNF/N-WASP-dependent epigenetic regulation of IL-23 that could explain how environmental stress affects IL-23 expression, thus contributing to chronic inflammatory diseases dependent on the IL-23/IL-17 axis.

For Supplementary Discussion see Supplementary Information.

## Methods

**Mice**. Mice with keratinocyte-restricted deletion of the N-WASP gene (N-WASP fl/fl K5 cre) were 129Sv/C56Bl6 outbred. The control mice used in all experiments are either N-WASP fl/fl or fl/+K5 cre (heterozygous ko), as we did not observe a difference between these two genotypes. In all experiments, age- and gender-matched litter mates were used as controls.

Mice with keratinocyte-restricted deletion of the IL23A gene were produced by CRISPR-Cas9-mediated insertion of loxP sites upstream of exon 2 and downstream of exon 4. Two single-guide RNAs (sgRNAs) were designed to target intron 1 and 4, respectively. Insertion of loxp sites was facilitated by 60 bp homologous arms identical to the genomic sequences flanking the cut sites (for sgRNA primers and homologous arms see Supplementary Table 2). The resulting mutant mice were than mated with N-WASP fl/fl K5 mice to generate mice with keratinocytes-restricted deletion of both N-WASP and IL-23a genes (dko).

Mice with keratinocyte-restricted deletion of the G9a gene were generated by crossing G9a fl/fl mice with K14-Cre mice (purchased from The Jackson Laboratory, USA), resulting in G9a conditional KO mice.

All mice were kept in an AAALAC (Association for Assessment and Accreditation of Laboratory Animal Care)-accredited animal house under specific pathogen-free conditions. Licenses for breeding and experimentation were obtained from the Danish Administration for Animal Experiments.

**Barrier assay**. To perform barrier assay, mouse E15.5 embryos, 2- and 6-day pups were killed and rinsed in phosphate-buffered saline (PBS), and then immersed in 25, 50, 75 100, 75, 50, and 25% ice-cold methanol for 2 min each. Subsequently, embryos and pups were immersed in 0.1% toluidine blue for 10 min on ice, afterwards washed in PBS and photographed.

**Cell isolation, culture and in vitro knockout**. Primary keratinocytes were isolated from skin adult mice by digestion with 0.8% trypsin in PBS for 30 min at 37 °C. Keratinocytes were cultured in minimum essential medium Eagle with 5 mg l$^{-1}$ insulin (Sigma #I5500), 10 µg l$^{-1}$ EGF (Sigma #E9644), 10 mg l$^{-1}$ transferrin (Sigma #T8158), 10 µM phosphoethanolamine (Sigma #P0503), 10 µM ethanolamine (Sigma #E0135), 0.36 µg l$^{-1}$ hydrocortisone (Calbiochem #386698), 1× GlutaMAX (Invitrogen #35050), 1× penicillin/streptomycin (Invitrogen #15140), and 40 ml of chelated fetal calf serum[32]. Normal human primary keratinocytes were cultured in Lonza KGM-2 BulletKit (CC-3107, Lonza, Swiss).

To introduce in vitro recombination, HEK293T cells (ATCC CRL-3216) were transfected with pRRLsin-GFP or pRRLsin-Cre-GFP plasmid together with pVSVg, pPAX-2. Next day, HEK293T were cultured in keratinocytes medium for 48 h to collect virus. Primary keratinocytes with fl/fl genotype were transduced with either green fluorescent protein (GFP) or Cre-GFP transducing virus for 48 h and analyzed after 5–7 days in culture.

**Inhibitor experiments**. For inhibitor experiments primary keratinocytes isolated from con and ko mice were treated for 24 h with 2 µM BIX01294 (3364, Tocris UK), 2 µM HC (Tranylcypromine hydrochloride; 3852, Tocris UK), 200 µM IOX-1 (4464, Tocris UK), 0.1 µM TSA (1406, Tocris UK), 50 µM CK548 (C7499, Sigma, USA), 0.5 µM latrunculin A (L5163, Sigma, USA), 10 µM MG-132 (1748, Tocris UK), or DMSO (Sigma, USA). For TNFα stimulation, keratinocytes were treated with 10 ng ml$^{-1}$ mouse TNFα (GF027, Millipore, USA) or as indicated in the legend for indicated times. For TNFα and IOX-1 experiment, keratinocytes were pretreated with 200 µM IOX-1 for 24 h and then followed by 4 h of TNFα stimulation. To prevent protein synthesis, primary keratinocytes were treated with 20 µg ml$^{-1}$ cycloheximide (46401, Sigma, USA) for indicated times. Normal human primary keratinocytes were treated as indicated in the legend.

**Subcellular fractionation**. Fractionation of primary keratinocytes was done as previously described[33,34]. Briefly, keratinocytes were collected by scraping, resuspened in buffer A (10 mM HEPES, pH 7.9, 10 mM KCl, 1.5 mM MgCl$_2$, 0.34 M sucrose, 10% glycerol, 1 mM dithiothreitol (DTT), 0.1% Triton X-100, protease and phosphatase inhibitors) for 10 min on ice, and centrifuged for 5 min at 1300 × g. The supernatant was cleared by centrifuging and collected as cytosol fraction (C). The pellet was the resuspended in buffer B (3 mM EDTA, 0.2 mM EGTA, 1 mM DTT, protease and phosphatase inhibitors) on ice for 30 min, and centrifuged for 5 min at 1800 × g. The supernatant was collect as soluble nuclear fraction (S), and the pellet was directly lysed as chromatin-enriched fraction (P). Alternatively, the pellet fraction was further extracted with buffer C (20 mM HEPES-KOH, 1.5 mM MgCl$_2$, 300 mM NaCl, 1 mM DTT, 0.2 mM EDTA, 20% (V/V) glycerol, pH 7.9) instead of buffer B for 30 min, and then centrifuged for 5 min at 1800 × g. The supernatant was collected as loose chromatin binding fraction (S300) and pellet as strong chromatin binding fraction (P300).

For micrococcal nuclease digestion, the pellet fraction was resuspended in buffer A with addition of 1 mM CaCl$_2$ and 2 U ml$^{-1}$ MNuclease (N5386, Sigma, USA) for 2 min at 37 °C. The sample was then centrifuged for 5 min at 1800 × g. The supernatant and pellet were collected and analyzed.

**Flow cytometry**. Preparative and analytical FACS was carried out according to standard protocols. For preparative FACS sorting, single-cell suspensions were prepared from epidermis[32], blocked by rat anti-CD16/32 (14-0161, eBiosciences, USA, 1:100 dilution) for 10 min on ice, and then stained with CD49f-FITC (555735, BD Pharmingen, USA, 1:100 dilution) and CD45.2-PE (12-0454, eBioscience, USA, 1:100 dilution) antibodies for 20 min on ice in dark. Samples were then sorted by FACSaria Cell sorter (BD Biosciences, USA). Sorted cells were lysed directly for RNA analysis.

Single-cell suspension from thymus, spleen, and bone marrow were stained with CD3ε-CY5.5 (100328, all from BioLegend, USA, 1:100 dilution), CD4-FITC (11-0041, 1:100 dilution), and CD8a-APC (17-0081, all from eBioscience, USA, 1:100 dilution).

For intracellular flow cytometry, single-cell suspensions were derived from epidermis as described above or from lymph nodes by pressing the tissue through a

**Fig. 9** TNF regulates *IL-23A* expression by phosphorylating N-WASP and degradation of G9a/GLP complex. **a** qRT-PCR analysis of IL-23A mRNA expression in con and ko primary keratinocytes treated for 4 h with TNFα (*n*: 3/3, mean ± SD, two-tailed unpaired *t*-test). **b** FACS analysis of surface TNF-RI in primary keratinocytes (*n*: 3/3, two-tailed unpaired *t*-test). **c** Primary keratinocytes were treated for indicated times with TNFα and then tested by immunoblot for expression of indicated proteins (*n*: 4/4). **d** Primary keratinocytes were treated for indicated time with TNF, fractionated, and analyzed by immunoblot for indicated proteins (C: cytosolic fraction, S300: soluble nuclear fraction extracted by 300 mM NaCl, P300: pellet after 300 mM NaCl extraction, SE: short exposure, LE: long exposure; *n*: 3). **e** ChIP of primary keratinocytes for H3K9me2 with qRT-PCR for indicated regions of IL-23A promoter after treatment with TNFα for indicated times (*n*: 5/5, mean ± SD, two-tailed unpaired *t*-test). **f** qRT-PCR analysis of *IL-23A* mRNA expression in con and ko primary keratinocytes pretreated with or without IOX-1 for 24 h and then stimulated with or without TNFα for 4 h as indicated. All normalized to GAPDH (*n*: 4/4, mean ± SD, two-tailed unpaired *t*-test; *$p \leq 0.05$; **$p \leq 0.01$; ***$p \leq 0.001$)

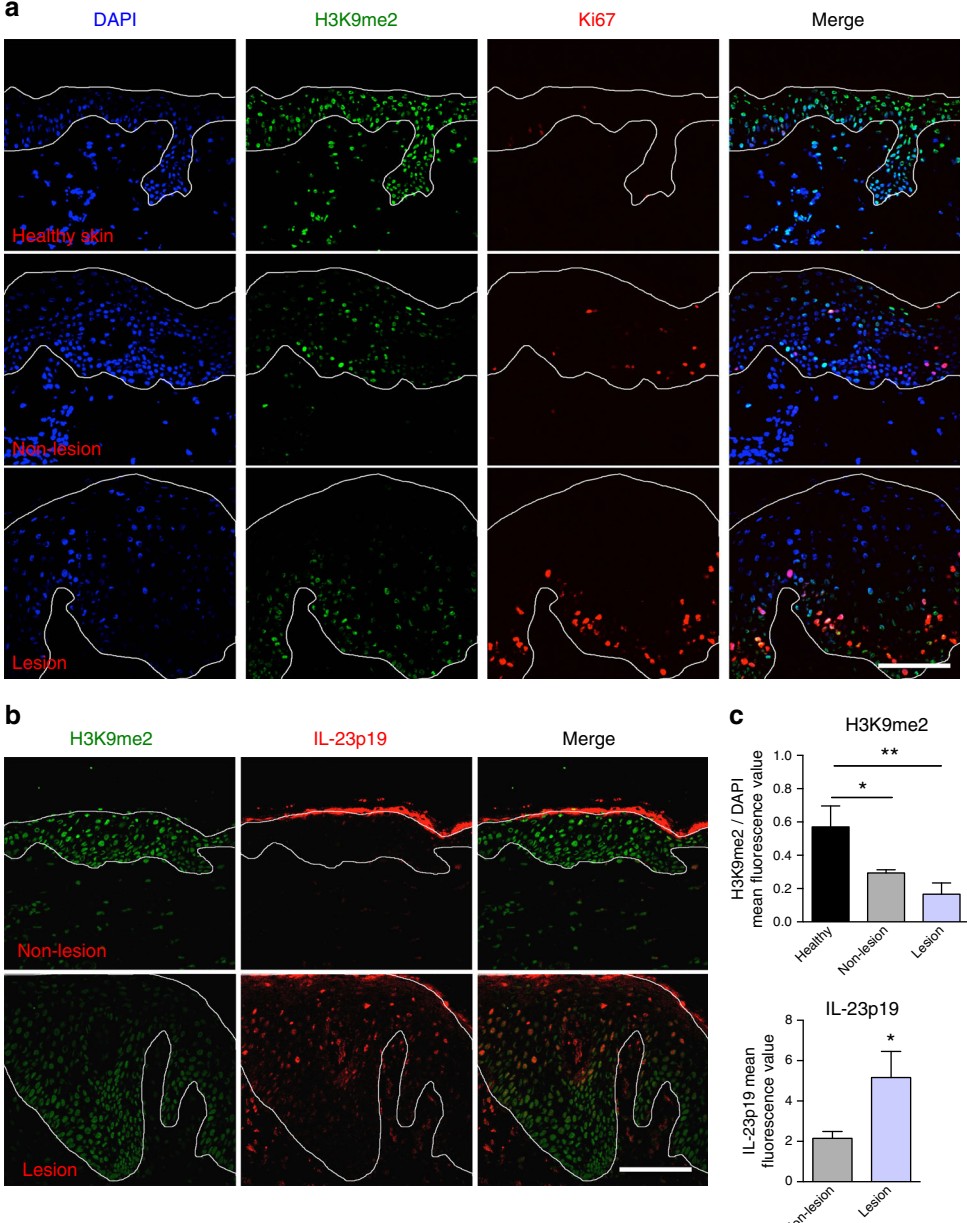

**Fig. 10** Decreased H3K9me2 in psoriatic lesions correlating with increased IL-23p19 expression. **a, b** H3K9me2 staining (**a**) and quantification (**b**) in healthy skin, non-lesioned and lesioned psoriatic skin (*n*: 3/3, mean ± SD, one-way ANOVA, with Tukey's multiple comparisons). **c** H3K9me2 and IL-23p19 staining in non-lesioned and lesioned psoriatic skin (*n*: 3/3). Scale bars: 100 μm (**a, c**). (*$p \leq = 0.05$; **$p \leq = 0.01$)

cell strainer (40 μm). To promote IL-17 expression and prevent secretion, cells were first stimulated for 6 h in RPMI containing cell stimulation cocktail plus protein transport inhibitors (00-4975-93, eBioscience, USA) at 37 °C. The cell suspension was then stained for surface markers for 20 min on ice in the dark, fixed with 4% paraformaldehyde for 10 min at room temperature, permeabilized with 0.1% Triton X-100 on ice for 2 min, blocked with blocking buffer on ice for 10 min and stained for intracellular markers. Data were collected by FACS Calibur (BD Biosciences, USA) using the CellQuestPro software and analyzed by Flowjo 8.8.2. The following antibodies were used: CD45.2-FITC (11-0454, 1:100 dilution), CD4-FITC (11-0041, 1:100 dilution), TCRγ–FITC (11-5711, 1:100 dilution), IL-17a-APC (17-7177-81, all eBioscience, USA, 1:100 dilution), TNFRI-PE (113003, 1:100 dilution), and CD3ε-CY5.5 (100328, 1:100 dilution, all BioLegend, USA).

**ChIP**. Chromatin Immunoprecipitation was done essentially as described in ref. [35]. Primary keratinocytes isolated from con and ko mice skin were grown to an approximate final count of $5 \times 10^7$–$1 \times 10^8$ cells for analysis. Cells were chemically crosslinked with 1% formaldehyde solution for 10 min at room temperature with gentle agitation and then quenched with 0.125 M glycine. Cells were rinsed three times with 1× PBS. Cells were resuspended, lysed, and sonicated to solubilize and shear crosslinked DNA. Sonication buffer (50 mM HEPES pH 7.5, 140 mM NaCl,

1 mM EDTA, 1% Triton X-100, 0.1% Na-deoxycholate, 0.1% SDS, and 2 mM PMSF, 1 μg ml$^{-1}$ leupeptin, 1 μg ml$^{-1}$ aprotinin). Bioruptor (Diagenode) was used to ensure consistent sonication between samples. Sonication was performed for 4–6 cycles (30 s-ON, 30 s-OFF). The resulting chromatin extract was precleared and appropriate amount was taken to incubate at 4 °C with 2–10 μg of H3K9me2 (4658, all from Cell Signaling, USA) antibody overnight. Next morning, pre-blocked beads were added and rotated for 3 h at 4 °C. Afterwards, beads were washed with sonication buffer twice, washing buffer (2 mM Tris pH 8.0, 0.02 mM EDTA, 50 mM LiCl, 0.1% NP-40 0.1% Na-deoxycholate) once, TE buffer once, followed by reverse crosslink for 5 h at 65 °C. DNA was then purified using the QIAGEN PCR purification kit according to the manufacturer's protocol. H3K9me2 relative occupancies were determined by calculating the immunoprecipitation efficiency (ratios of the amount of immunoprecipitated DNA to that of the input sample). Antibodies against H3K9me2 (4658, Cell Signaling, USA) were used for immunoprecipitation. Primers used for qRT-PCR of ChIP are listed in Supplementary Table 2.

**Gene expression analysis**. RNA from skin and keratinocytes culture was prepared using the GeneElute Mammalian Total RNA miniprep kit (RNT350, Sigma, USA) according to the instructions of the manufacturer. Skin samples for RNA were first

kept in RNA later (R0901, Sigma) until isolation. RNA was reverse transcribed to complementary DNA (cDNA) by RevertAid H Minus First Strand cDNA Synthesis Kit (K1632, Thermo Scientific, USA). qRT-PCR was performed on the Applied Biosystems 7300 Real Time PCR system using SYBR green (K0221, Thermo Scientific, USA) incorporation following standard protocols. The Ct value was calculated based on duplicates and normalized to the housekeeping gene GAPDH. Primers used for gene expression analysis are listed in Supplementary Table 2.

Microarray analysis was carried out at the Copenhagen University Hospital Microarray Center. Total RNA samples were isolated from total back skin of 3 pairs of 7-week-old control and mutant mice. RNA samples in each group were mixed at equal amounts and transferred to the Copenhagen University Hospital Microarray Center, where labeling of probes, hybridization to GeneChip MouseGene ST 1.0 (Affymetrix), and scanning of the arrays was carried out.

**Immunofluorescence and immunohistochemistry**. Cryo and paraffin sections of mice as well as primary keratinocytes cultured on glass slides were analyzed by immunofluorescent, immunohistochemical, and histochemical staining as described previously[18]. Autoimmune antibodies in serum were detected by incubating mouse keratinocytes with serum (1:100) and then with Cy-3-conjugated goat anti-mouse. The following antibodies were used: CD45.2 (11-0454, eBioscience, USA, 1:100 dilution), Cre (15036S, Cell Signaling, 1:100 dilution), keratin 6 (PRB-169p, Covance, USA, 1:400 dilution), H3K9me2 (4658, Cell Signaling, USA, or ab1220, Abcam, UK, 1:200 dilution), IL-23p19 (LS-B573, LSBio, USA, 1:400 dilution), IL-23p19 (511201, Biolegend, USA, 1:100 dilution), Ki67 (ab15580, Abcam, UK, 1:400 dilution), and CD49f-FITC (555735, BD Pharmingen, USA, 1:100 dilution). As secondary reagent, 488- or Cy-3-conjugated goat anti-rabbit and goat anti-mouse IgGs were used (both from Jackson ImmunoResearch, USA). Nuclear counter-staining was performed with DAPI (4′,6-diamidino-2-phenylindole; Sigma, USA). Mast cells were stained with toluidine blue (89640, Sigma, USA) and F-actin with Alexa Fluor 488 phalloidin (A12379, Invitrogen, USA, 1:400 dilution). Images were analyzed with a Leica DM RXA2 confocal microscope, equipped with 20× HC PL Apo (NA 0.70), 40× HCX PL APO (NA 1.25-0.75), and 63× HCX PL APO (NA 1.40-0.60) objectives and controlled by Leica Microsystems confocal software (version 2.61 Build 1537; all from Leica Microsystems). ImageJ was used to quantify fluorescent images.

**Human skin samples**. Biopsies of healthy skin were obtained from 3 male donors (ages 24, 46, and 51 years) and biopsies of lesional and corresponding non-lesional skin were collected from 3 male patients with plaque psoriasis (ages 24, 25, and 57 years) under local ethical permissions and upon written informed consent. Cryo sections of the human skin samples were prepared and stained as described above.

**Biochemical analysis**. Western blotting was performed according to standard protocols. The following antibodies were used: pSmad2 (S465, S467) (3108S, 1:1000 dilution), H3K9me2 (4658, 1:4000 dilution), N-WASP (4848, 1:2000 dilution), pSAPK/JNK (T183, Y185, 1:1000 dilution) (9251), pIRAK4 (T345, S346) (11927S, 1:1000 dilution), pNF-κB p65 (S536) (3033, 1:1000 dilution, all from Cell Signaling, USA), G9a (3306, 1:500 dilution, Cell Signaling, USA, or 07-551, 1:500 dilution, Millipore, USA), PCNA (ab29, 1:5000 dilution), β-actin (ac15, 1:10,000 dilution), GLP (ab41969, 1:500 dilution all from Abcam, UK), Histone H3 (sc-8654, 1:2000 dilution), GAPDH (sc-25778, 1:5000 dilution), pp38 Thr180/Tyr182 (sc-17852, 1:2000 dilution, all from Santa Cruz, USA), and pN-WASP (S480, S481) (MBS472040, 1:1000 dilution MyBioSource, USA).

As secondary agents, horseradish peroxidase (HRP)-coupled goat anti-rabbit, goat anti-mouse, and donkey anti-goat antibodies were used (all from Jackson ImmunoResearch, USA). All results were quantified using TotalLab TL100 software (Nonlinear Dynamics). GAPDH, PCNA, β-actin, or Histone H3 was used for normalization.

Nuclear co-immunoprecipitation (co-IP) was done using the Nuclear Complex Co-IP Kit (54001, Active Motif, USA) according to the manufacturer's instruction. A total of 50 μl Protein A agarose (16-125, Millipore) was used to pull down the antibodies. Antibodies used for co-IP were N-WASP (4848, 1:50 dilution) and G9a (3306, 1:50 dilution all from Cell Signaling, USA).

For co-IP of IL-23 heterodimer, the epidermal cells were lysed in RIPA buffer for 30 min on ice and cleaned with Protein A agarose for 1 h at 4 °C. Rat anti-mouse IL-23p19 (16-7232-81, Invitrogen, USA, 1:50 dilution) was added to the cleaned lysate and precipitated by Protein A agarose. Rabbit anti-mouse IL-23p19 (NBP1-77257, Novus, USA, 1:1000 dilution) and rat anti-mouse IL-12p40 (505201, BioLegend, USA, 1:100 dilution) were used for detection.

Uncropped scans of the most important blots are shown in Supplementary Figure 4.

**ELISA**. Tissues protein lysates were obtained by using a Precellys® 24 bead-based homogenizer (Precellys® Bertin Technologies, France) that allowed rapid protein extraction of multiple samples. In brief, tissues were homogenized in cold 1× Cell lysis buffer (Cell Signalling Technology, Danvers, MA, USA) containing a freshly added protease inhibitor cocktail (Roche Applied Science, Penzberg, Germany) and phosphatases inhibitor cocktail (Thermo Fischer Scientific, Walldorf, Germany), followed by incubation in ice for 20 min to allow a complete lysis. Then, cell debris was removed by centrifugation at 15,000 × g for 15 min at 4 °C and the

supernatants were transferred to a fresh microcentrifuge tube. Total protein concentration of all the samples was determined by BCA protein assay (Thermo Fischer Scientific, USA). Ultimately, all samples were normalized to the same total protein concentration of 2 mg ml$^{-1}$, prior analysis for the presence of the protein of interest.

IL-1β, IL-12p70, and mKC were analyzed using the Th1/Th2 panel multiplex kit from Meso Scale Discovery (MSD, Gaithersburg, MD, USA) and read on a Sector Imager 6000 (MSD). IL-17F was analyzed using ELISA kit from R&D (R&D Systems, Abingdon, UK). All assays were used following the instruction of the manufacturer.

For anti-dsDNA ELISA, 100 μl of 5 μg ml$^{-1}$ dsDNA isolated from mouse tail was bound to microtiter plate at 4 °C overnight. The plate was blocked with 1% bovine serum albumin (BSA) in PBS for 90 min, then incubated with 1% BSA in PBS (negative control), anti-dsDNA antibody (1:200; MAB1293, Milipore, USA; positive control), and sera from con and ko mice (1:200) for 1 h. Sera and antibodies were diluted in 1% BSA in PBS. After washing three times with 0.1% Tween-20 (Sigma) in PBS (PBST), the plate was incubated with anti-mouse IgG HRP secondary antibody (1:1000; Jackson Immunoresearch) for 1 h and afterwards washed three times with PBST. Plate-bound anti-dsDNA antibodies were detected by 100 μl 3′,3′,5′,5′-tetramethyl benzidine substrate solution (T4444, Sigma-Aldrich). The reaction was then transferred to a new plate and read at 650 nm on a microplate reader (Glomax-multi+ detection system, Promega).

**Statistics**. Data are presented as means with error bars representing standard deviation (SD). Statistical significance was determined either by the two-tailed Student's $t$-test, or by one-way analysis of variance (ANOVA), with Tukey's multiple comparisons if not stated otherwise. Significant differences are indicated by asterisks (*$p \leq 0.05$; **$p \leq 0.01$; ***$p \leq 0.001$; ****$p \leq 0.0001$).

**Data availability**. Microarray data are available at ArrayExpress with accession ID E-MTAB-6580 (https://www.ebi.ac.uk/arrayexpress/experiments/E-MTAB-6580/). All other data are available from the corresponding author upon request.

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

## Acknowledgements

We thank Mrs. Anna Fossum and Mr Volkan Turan for excellent technical help and the Danish Medical Research Council and the LEO Foundation for support.

## Author contributions

H.L. designed and conducted experiments and wrote the manuscript. Q.Y., A.G.M., X. W., J.H., P.L., H.N., M.S.M., L.V., and B.R. conducted experiments. A.W., C.V., S.F.T., A. A., and S.A.B. provided crucial reagents. E.P. and K.H. provided important advice. C.B. designed experiments and wrote the manuscript.

## Additional information

**Competing interests:** The authors declare no competing interests.

