## [Peer Review File(PDF 271 kb) · Nature Communications]

Reviewers' comments:

Reviewer #1 (Remarks to the Author):

The Manuscript "Epigenetic control of IL-23 expression in keratinocytes is important for chronic skin inflammation" is both interesting but confusing in a few parts. Here are a few suggestions. 1-4 are more general and 5-8 are more specific about the data in the figures.

1) The authors state that "in keratinocytes of psoriatic lesions a decrease in H3K9 dimethylation precedes IL-23 expression, suggesting relevance for disease development" As far as I can tell, the authors base this claim on descriptively comparing the expression level by IF in tissue sections of H3K9 methyl and IL23 between lesional and non-lesional skin. Non-lesional skin is not the same as pre-lesional skin and cannot be compared as such. On the contrary psoriatic lesions often re-develop in previous lesional rather than non-lesional skin.

2) There is conflicting data on the role of N-WASP deletion. The authors of this paper have previously (Lefever et al. J Cell Sci 2010 123: 128-140; doi: 10.1242/jcs.053835) demonstrated that keratinocyte deletion of N-WASP –leads to reduced keratinocyte proliferation in vitro "WASP-deficient keratinocytes grew slower than control keratinocytes, and displayed a significant increase of cells in the G1 phase, indicating a defect in G1-S transition (Fig. 8A,B). This growth defect was not caused by impaired adhesion, increased apoptosis or premature differentiation of N-WASP-null keratinocytes (supplementary material Fig. S5B,C). These results indicate that primary N-WASP-null keratinocytes have an impaired proliferation"

In the paper by Lyubimova et al. JCI doi 10.1172/JCI36478) led to severe alopecia, epidermal hyperproliferation, and ulceration, without obvious effects on epidermal differentiation and wound healing, and gradual stem cell depletion.

This phenotype does not highly resemble psoriasis. Why are the authors using psoriasis as an example for the signaling pathways induced by N-WASP deletion? Have the authors compared cytokines elevated in atopic dermatitis, alopecia, contact dermatitis?

3) The authors explain the discrepancies that they tested with only few mice and that not detecting K6 previously may be due to differences in genetic background and different animal housing. Can the authors compare these different genetic backgrounds and animal housing conditions side by side to demonstrate this?

4) What is the relevance for psoriasis for comparing IgG deposition and levels? Same with the kidney phenotype?

5) Figure 3D: Is this quantification significant using a two-tailed for IL23p19 and n=3 per group?

6) Figure 7F: Is this quantification significant using a two-tailed unpaired t.test for H3K9me2 and n=4 per group? What does ** indicate?

7) Figure 9C: The immunoblots with GLP and GAPDH have a white (no) background

8) Figure 9E: Is this quantification significant using a two-tailed unpaired t.test for H3K9me2 and n=4 per group? What does * indicate?

Reviewer #2 (Remarks to the Author):

This is an excellent manuscript describing the importance of keratinocyte-produced IL-23 in

causing a chronic skin inflammation. The authors uncovered a novel molecular pathway that mediates TNF α -controlled expression of IL-23 via previously uncharacterised nuclear function of N-WASP by regulating the degradation of the histone methyltransferases G9a and GLP, and H3K9 dimethylation of the IL-23 promoter. The authors also showed the relevance of the decreased H3K9 dimethylation in deregulation of IL-23 expression in psoriatic lesions. The results and conclusions are made based on solid methodology and comprehensive analysis.

A few minor comments should be addressed.

1. Page 9, line 9: Fig. 6D should refer to Fig. 3D
2. Page 10, line 11: Fig. 3A should refer to Fig. 3E
3. Page 12: Authors claim that IL-19 expression was not altered in dko compared to ko, despite of a significant increase in IL-19 expression as shown on Fig. 5D.
4. Page 12: the authors claim that "average expression of IL-17F, IL-19, and LL37 was decreased, although the changes were not significant" in FACS purified keratinocytes. However, Fig. 6A does not show any changes in expression of these cytokines.
5. Please provide a legend for Fig 6G.
6. The authors provide no evidence that IOX-1 treatment reduces IL-23 expression in both control and N-WASP ko keratinocytes: Fig. 7B only demonstrates that IOX-1 treatment increases H3K9me2 levels.

Reviewer #3 (Remarks to the Author):

I read with interest this manuscript by Dr. Li et al on epigenetic control of IL-23 expression in keratinocytes and its role in chronic skin inflammation. I have the following comments.

1. The authors demonstrate that loss of N-WASP in mouse keratinocytes lead to chronic skin inflammation. One limitation that I kept coming back at was that while the authors show increased IL-23p19 mRNA expression and anti-p19 positive staining in the epidermis they never truly demonstrate increased IL-23 protein expression and secretion by keratinocytes in any of the experiments presented in this manuscript – this is very surprising to me. Overall the whole data presented in this manuscript is based on the assumption that the IL-23 protein is increased because IL23A mRNA expression is increased as well as the decreased phenotype of the N-wasp/IL23a double-knockout. This should be a relatively easy experiment for the authors to address and I feel that it should be included.

2. It is not clear why the authors only show mRNA expression for 15 chemokines and cytokine in Figure 3A. Given the findings that correlate with lupus such as the autoantibody formation it would be interesting to determine expression of cytokines such as type I IFNs – particularly as CXCL10 is one of the more robustly elevated chemokines early in this model (day 6) and it typically not a chemokine induced by IL-23 but more characteristically type I or type II IFNs. Likewise it is not clear why IL12B and IL12A expression is not shown.

3. Its confusing to me why the authors measure variable markers at different timepoints, i.e. in Figure 1C, E, and D they show data for 6d, 9d, 3w and 7w, in Figure 1F only 2w and 4w data are shown, in Figure 1G, 9d, 2w, 3w, 5w and 7w data are shown. In figure 2A it is 3w and 7 w, Fig 2C It is 2w, 4w and 8w. Figure 3A it is 6d and 7w but in Figure 3D it is 6d and 4w. Figure 5 it is 6-8wk. This continues throughout the whole manuscript and given the robustness of the phenotype and what appears to be a continually worsening phenotype (based on Figure 1D) it would appear to me that the later timepoints would be most relevant....therefore why some of the figures are limited to data from the first 4 weeks is not clear.

4. The psoriasis data presented in figure 10 is very weak, correlative in nature, and is based on

immunofluorescence alone. The phenotype of the N-wasp mouse model is not compatible with psoriasis given presence of autoantibodies in serum and binding in the skin. While the IL-23 mechanism might be relevant to psoriasis the authors should take steps to make this much more robust – i.e. by using ex vivo psoriatic keratinocyte cultures or do ChIP-pcr of psoriatic skin.

5. There is no mention of the background mouse strain used for these studies. This should be clear. Also, the authors should consider using mouse genetic nomenclature for protein and gene expression.

Minor points

1. Figure 1B. The images of the control and KO tail are blurry and should be replaced with clearer images

2. Figure 6A. It is very surprising to me that the authors can detect IL-17F expression in purified keratinocytes in vivo at fairly high levels. Is this correct? Why isn't IL-17A included in this figure (I'd expect this to be absent as well).

Response to Reviewers' comments:

Reviewer #1 (Remarks to the Author):

The Manuscript "Epigenetic control of IL-23 expression in keratinocytes is important for chronic skin inflammation" is both interesting but confusing in a few parts. Here are a few suggestions. 1-4 are more general and 5-8 are more specific about the data in the figures.

1) The authors state that "in keratinocytes of psoriatic lesions a decrease in H3K9 dimethylation precedes IL-23 expression, suggesting relevance for disease development" As far as I can tell, the authors base this claim on descriptively comparing the expression level by IF in tissue sections of H3K9 methyl and IL23 between lesional and non-lesional skin. Non-lesional skin is not the same as pre-lesional skin and cannot be compared as such. On the contrary psoriatic lesions often re-develop in previous lesional rather than non-lesional skin.

We removed this speculation from the text.

To support that a H3K9me2 dependent regulation of IL-23 expression takes place also in human keratinocytes we introduced to the revised manuscript now data with primary human keratinocytes. Treatment of primary human keratinocytes cells for 30 min with TNF significantly decreased G9a and H3K9me2 levels, while it increased NFkB phosphorylation (Suppl. Fig. 3E). Treatment with the G9a/GLP inhibitor BIX for 24h decreased H3K9me2 levels and increased IL-23A expression in a concentration dependent manner (Suppl. Fig. 3A). On the other hand, H3K9me2 levels were increased and IL-23A mRNA levels decreased by incubation for 24h with the histone-demethylase inhibitor IOX-1 (Suppl. Fig. 3C). Importantly, TNF induced IL-23A expression was completely blocked by IOX-1, while BIX treatment increased TNF induced IL-23A expression (Suppl. Fig. 3B, D).

These data demonstrate that also in human keratinocytes H3K9me2 levels are crucially important for regulating basal and TNF induced IL-23A expression.

2) There is conflicting data on the role of N-WASP deletion. The authors of this paper have previously (Lefever et al. J Cell Sci 2010 123: 128-140; doi: 10.1242/jcs.053835) demonstrated that keratinocyte deletion of N-WASP –leads to reduced keratinocyte proliferation in vitro "WASP-deficient keratinocytes grew slower than control keratinocytes, and displayed a significant increase of cells in the G1 phase, indicating a defect in G1-S transition (Fig. 8A,B). This growth defect was not caused by impaired adhesion, increased apoptosis or premature differentiation of N-WASP-null keratinocytes (supplementary material Fig. S5B,C). These results indicate that primary N-WASP-null keratinocytes have an impaired proliferation"

In the paper by Lyubimova et al. JCI doi 10.1172/JCI36478) led to severe alopecia, epidermal hyperproliferation, and ulceration, without obvious effects on epidermal differentiation and wound healing, and gradual stem cell depletion.

This phenotype does not highly resemble psoriasis. Why are the authors using psoriasis as an

example for the signaling pathways induced by N-WASP deletion? Have the authors compared cytokines elevated in atopic dermatitis, alopecia, contact dermatitis?

The skin inflammation is a phenotypic aspect that was not described before in mice with a keratinocyte-restricted deletion of N-WASP. It is not in conflict to the earlier reported reduced proliferation of primary N-WASP null keratinocytes in vitro (Lefever et al., 2010), which we still observe and now understand also mechanistically (manuscript in preparation). This novel mechanism also explains the gradual reduction of keratinocyte stem cells in vivo, first described by Lyubimova and colleagues.

To investigate the skin inflammation observed in N-WASP mutant mice, we tested a number of cytokines described to be involved in different inflammatory processes of the skin such as atopic dermatitis, irritant contact dermatitis alopecia areata, and psoriasis (Suarez-Farinas et al., 2015; Lee et al, 2013). The cytokine profiles for these diseases are highly overlapping, but the IL-23/IL-17A axis seems to be particularly increased in psoriasis (Suarez-Farinas et al., 2015). Furthermore, antibodies against IL-23 are in clinical use for psoriasis, but up to now not for any other skin inflammation, underlining the importance of IL-23/IL-17A particularly for psoriasis.

We therefore focused our investigation on understanding the molecular mechanism for the increased expression of IL-23 in N-WASP-null keratinocytes, although also other mediators are elevated in skin of 6d old mutant mice in vitro and primary N-WASP-null keratinocytes in vitro.

The identified H3K9 methylation dependent regulation of IL-23 expression might of course also be of relevance for other skin inflammatory diseases than psoriasis and altered H3K9 methylation might affect expression of other inflammation related mediators than IL-23.

N-WASP mutant mice therefore model a mechanistic pathway of potential relevance for skin inflammations such as psoriasis, but they do not display all clinical aspects of psoriasis, indicating that other mechanistic pathways important for human psoriasis are not modelled sufficiently. They furthermore show phenotypic aspects such as autoimmune antibodies that are not related to psoriasis but similar to other diseases, which will be interesting to follow up in future studies.

We changed the manuscript to make these points clear and to avoid misinterpretation of our mechanistic findings.

3) The authors explain the discrepancies that they tested with only few mice and that not detecting K6 previously may be due to differences in genetic background and different animal housing. Can the authors compare these different genetic backgrounds and animal housing conditions side by side to demonstrate this?

The two studies are separated by about 6-7 years of breeding of a small number of mice and the animal houses in question have partially changed with respect to cages (open cages vs. ivc), bedding material, ventilation, etc. during that time. It is therefore unfortunately not possible to make a direct comparison at this time point to solve this question. K6 staining has been in our hand always a very reliable staining with an internal control on each section by the staining of K6 in the companion layer

of the hair follicle. We therefore think that changes in genetic background and animal house environment are the most likely explanations for the observed phenotypic difference, but we cannot prove it.

4) What is the relevance for psoriasis for comparing IgG deposition and levels? Same with the kidney phenotype?

As mentioned above, N-WASP mutant mice show phenotypic aspects that are not observed in psoriasis and both hydronephrosis and IgG deposition are examples for that. Our hypothesis for the kidney phenotype is that loss of N-WASP in the ureter epithelium is leading to local inflammation, occlusion and consequently hydronephrosis. Such a mechanism would be clearly of no relevance in psoriasis, where no loss of N-WASP gene function or expression has been reported in ureter epithelium.

We discuss this in the revised manuscript.

5) Figure 3D: Is this quantification significant using a two-tailed for IL23p19 and n=3 per group?

Yes, it was significant using a two-tailed t-test.

6) Figure 7F: Is this quantification significant using a two-tailed unpaired t.test for H3K9me2 and n=4 per group? What does ** indicate?

No, it was significant using a paired two-tailed t test. We now made 3 additional experiments and present a new figure 7F, where the difference is significant with a two tailed, unpaired t-test. ** indicate $p < 0.01$ as described in the "Experimental Procedures" under the chapter "Statistics".

7) Figure 9C: The immunoblots with GLP and GAPDH have a white (no) background

We exchanged this image. The new Fig. 9C now includes the background.

8) Figure 9E: Is this quantification significant using a two-tailed unpaired t.test for H3K9me2 and n=4 per group? What does * indicate?

No, it was significant using a paired two-tailed t test. We now made 1 additional experiment and present a new figure 9E, where the difference is significant with a two tailed, unpaired t-test. * indicates $p < 0.05$ as described in the "Experimental Procedures" under the chapter "Statistics".

Reviewer #2 (Remarks to the Author):

This is an excellent manuscript describing the importance of keratinocyte-produced IL-23 in causing a chronic skin inflammation. The authors uncovered a novel molecular pathway that mediates TNF α -controlled expression of IL-23 via previously uncharacterised nuclear function of N-WASP by regulating the degradation of the histone methyltransferases G9a and GLP, and H3K9 dimethylation of the IL-23 promoter. The authors also showed the relevance of the decreased H3K9 dimethylation in deregulation of IL-23 expression in psoriatic lesions. The results and conclusions are made based on solid methodology and comprehensive analysis.

A few minor comments should be addressed.

1. Page 9, line 9: Fig. 6D should refer to Fig. 3D

We corrected this wrong figure number. The mRNA values in total skin for IL-23A and IL-12B are shown in Fig. 5D. IF staining for IL-23p19 protein is shown in Fig. 3D. The revised version now contains also a Western blot for IL-23 (heterodimer of IL-23p19 + IL12p40) and IL-12p40 (Fig. 5E).

2. Page 10, line 11: Fig. 3A should refer to Fig. 3E

We corrected this mistake.

3. Page 12: Authors claim that IL-19 expression was not altered in dko compared to ko, despite of a significant increase in IL-19 expression as shown on Fig. 5D.

We corrected this mistake. IL-19 is now mentioned among the genes upregulated in dko compared to ko.

4. Page 12: the authors claim that “average expression of IL-17F, IL-19, and LL37 was decreased, although the changes were not significant” in FACS purified keratinocytes. However, Fig. 6A does not show any changes in expression of these cytokines.

In the old version of the manuscript we did not describe, which mice are compared in this statement. In the revised version, we therefore changed the sentence to: Interestingly, average

expression of IL-17F, IL-19, and LL37 was decreased in both ko and dko keratinocytes compared to controls, although the changes were not significant.

5. Please provide a legend for Fig 6G.

We added a description of the right panel to the figure legend of the revised manuscript: The right panel shows the same data, but normalized to the respective DMSO treated samples, thus showing the relative effect of the treatment as fold change.

6. The authors provide no evidence that IOX-1 treatment reduces IL-23 expression in both control and N-WASP ko keratinocytes: Fig. 7B only demonstrates that IOX-1 treatment increases H3K9me2 levels.

A reference to Fig. 6G was missing in the text, which we added now. Fig. 6G shows that IOX-1 treatment reduces IL-23A expression in control and N-WASP ko keratinocytes.

Reviewer #3 (Remarks to the Author):

I read with interest this manuscript by Dr. Li et al on epigenetic control of IL-23 expression in keratinocytes and its role in chronic skin inflammation. I have the following comments.

1. The authors demonstrate that loss of N-WASP in mouse keratinocytes lead to chronic skin inflammation. One limitation that I kept coming back at was that while the authors show increased IL-23p19 mRNA expression and anti-p19 positive staining in the epidermis they never truly demonstrate increased IL-23 protein expression and secretion by keratinocytes in any of the experiments presented in this manuscript – this is very surprising to me. Overall the whole data presented in this manuscript is based on the assumption that the IL-23 protein is increased because IL23A mRNA expression is increased as well as the decreased phenotype of the N-wasp/IL23a double-knockout. This should be a relatively easy experiment for the authors to address and I feel that it should be included.

We show in the revised Fig. 3E now a significantly increased expression of IL-23p19 and IL12p40 protein in lysates of N-WASP ko mice compared to controls (Fig. 3E). We furthermore immunoprecipitated the IL-23 heterodimer from epidermal lysates by antibodies against IL-23p19 (rat anti mouse) and detected the IL-23 subunits by an antibody against IL-12p40 and against IL-23p19 (rabbit anti mouse). Quantification indicated a significant increase in IL-23 heterodimer in the epidermis of N-WASP ko mice (Fig. 3F).

2. It is not clear why the authors only show mRNA expression for 15 chemokines and cytokine in Figure 3A. Given the findings that correlate with lupus such as the autoantibody formation it would be interesting to determine expression of cytokines such as type I IFNs – particularly as CXCL10 is one of the more robustly elevated chemokines early in this model (day 6) and it typically not a chemokine induced by IL-23 but more characteristically type I or type II IFNs. Likewise it is not clear why IL12B and IL12A expression is not shown.

Testing IFN α and IFN β mRNA expression in total skin, we found both upregulated in N-WASP ko mice older than 7 w. In 6d old mice, however, neither of them is increased. We added these data to a revised Fig. 3A.

IL-12A expression is shown in Fig. 3A as “IL-12” and both IL-12A and IL-12B are shown in Fig. 5D as “IL-12” and “IL-12p40”. To avoid confusion, we changed in the revised version to IL-12A and IL-12B as names for the genes and IL-12p40 as name for the protein of IL-12B. Similarly, we used IL-23p19 only for the protein and IL-23A for the corresponding gene.

3. Its confusing to me why the authors measure variable markers at different timepoints, i.e. in Figure 1C, E, and D they show data for 6d, 9d, 3w and 7w, in Figure 1F only 2w and 4w data are shown, in Figure 1G, 9d, 2w, 3w, 5w and 7w data are shown. In figure 2A it is 3w and 7 w, Fig 2C It is 2w, 4w and 8w. Figure 3A it is 6d and 7w but in Figure 3D it is 6d and 4w. Figure 5 it is 6-8wk. This continues throughout the whole manuscript and given the robustness of the phenotype and what appears to be a continually worsening phenotype (based on Figure 1D) it would appear to me that the later timepoints would be most relevant....therefore why some of the figures are limited to data from the first 4 weeks is not clear.

The major reason for the different time points used in our study was the age of the mice available to be tested in parallel for the respective experiments. The use of exact similar time points for all experiments would have required significantly increased breeding which was unfortunately beyond our budget possibilities. Another reason for different time points used is that during the initial analysis we were not clear about the phenotype and the corresponding most relevant time points. With hindsight, we could have done it more efficiently.

4. The psoriasis data presented in figure 10 is very weak, correlative in nature, and is based on immunofluorescence alone. The phenotype of the N-wasp mouse model is not compatible with psoriasis given presence of autoantibodies in serum and binding in the skin. While the IL-23 mechanism might be relevant to psoriasis the authors should take steps to make this much more robust – i.e. by using ex vivo psoriatic keratinocyte cultures or do ChIP-pcr of psoriatic skin.

We agree completely with the reviewer that the presence of autoantibodies in N-WASP ko mice is not compatible with psoriasis. Absence of a strong lymphocytic infiltrate is another major difference to psoriasis. We mention these differences in the manuscript. However, the decreased amounts of H3K9me2 correlating with increased IL-23 is observed in N-WASP mutant mice as well as in lesions of psoriatic patients, suggesting that this H3K9me2 dependent regulation of IL-23 is occurring in humans and might contribute to the disease, although definitely not explaining it completely. This novel mechanism might also be involved in increased IL-23 expression in other diseases, which should be tested in further studies. We tried to make these points more clear in the revised version of the manuscript.

The suggested experiments of testing ex vivo psoriatic keratinocytes cultures or performing CHIP-PCR of psoriatic skin are problematic for different reasons.

H3K9me2 is a highly regulated mark in keratinocytes and strongly dependent on the environmental conditions, as we show in our study. Treatment of control keratinocytes with mediators such as TNF results in rapid, but only transient changes of H3K9me2 and IL-23A expression. Plating and expansion of psoriatic keratinocytes will therefore result in a normalization of H3K9me2 and IL-23A expression.

CHIP-PCR from lysates of psoriatic lesions would circumvent the problem of culturing, but is technically very challenging, since it has to be done from lysates of epidermis derived from 3 mm punch biopsies. The corresponding protein amounts are too low for standard CHIP-PCR protocols. We therefore tested a published microCHIP protocol (Dahl and Collas, 2009) with different amounts (5000 cells, 20000 cells) of primary human keratinocytes using 4 set of primers spanning the 1 kB region upstream of the transcription start site of the human IL23A Gene. However, in none of the experiments we could detect a signal.

We, therefore, did not proceed with patient samples although we had established already a corresponding clinical collaboration and obtained an ethical permit for the experiments from the authorities.

As an alternative way to check whether a TNF/H3K9me2/IL-23A pathway might be of relevance in human keratinocytes, we tested in the revised version primary human keratinocytes in vitro for IL-23A expression in response to TNF and to alteration of H3K9me2 levels.

Treatment of primary human keratinocytes cells for 30 min with TNF significantly decreased G9a and H3K9me2 levels, while it increased NFkB phosphorylation (Suppl. Fig. 3E). Treatment with the G9a/GLP inhibitor BIX for 24h decreased H3K9me2 levels and increased IL-23A expression in a concentration dependent manner (Suppl. Fig. 3A). On the other hand, H3K9me2 levels were increased and IL-23A mRNA levels decreased by incubation for 24h with the histone-demethylase inhibitor IOX-1 (Suppl. Fig. 3C). Importantly, TNF induced IL-23A expression was completely blocked by IOX-1, while BIX treatment increased TNF induced IL-23A expression (Suppl. Fig. 3B, D).

These data demonstrate that also in human keratinocytes H3K9me2 levels are crucially important for regulating basal and TNF induced IL-23A expression.

5. There is no mention of the background mouse strain used for these studies. This should be clear. Also, the authors should consider using mouse genetic nomenclature for protein and gene expression.

The genetic background was mentioned and referenced in the Method section. We rephrased now the description of the mice stressing the genetic background in the revised version of the manuscript.

We changed in the revised version to IL-12A and IL-12B as names for the genes and IL-12p40 as name for the protein of IL-12B. Similarly, we used IL-23p19 only for the protein and IL-23A for the gene.

Minor points

1. Figure 1B. The images of the control and KO tail are blurry and should be replaced with clearer images

We replaced the old Fig. 1B by a new figure with higher resolution.

2. Figure 6A. It is very surprising to me that the authors can detect IL-17F expression in purified keratinocytes in vivo at fairly high levels. Is this correct? Why isn't IL-17A included in this figure (I'd expect this to be absent as well).

Indeed, we also found expression of IL-17F in primary keratinocytes (Fig. 6A) surprising and therefore rechecked our primer sequences several times, to assure that there is no mistake. Interestingly, IL-17F expression in total skin was significantly higher (Fig. 5D) than in primary keratinocytes and was strongly increased in ko and dko, while it was decreased in isolated ko and dko keratinocytes compared to control. This suggests that keratinocytes are only a very minor source of IL-17F.

IL-17A expression was basically absent in isolated keratinocytes purified by FACS as shown in Fig. 3E. We tested it again in the experiment presented in Fig. 6A, but again could not detect it and therefore did not show the data in that figure (As with IL-22 and IFN γ). We describe these findings in the revised version as "data not shown".

References:

Dahl JA, Collas P. A rapid micro chromatin immunoprecipitation assay (microChIP). Nat Protoc. 2008;3(6):1032-45

Suárez-Fariñas M, Ungar B, Noda S, Shroff A, Mansouri Y, Fuentes-Duculan J, Czernik A, Zheng X, Estrada YD, Xu H, Peng X, Shemer A, Krueger JG, Lebwohl MG, Guttman-Yassky E. Alopecia areata profiling shows TH1, TH2, and IL-23 cytokine activation without parallel TH17/TH22 skewing. *J Allergy Clin Immunol*. 2015 Nov;136(5):1277-87.

Lee HY, Stieger M, Yawalkar N, Kakeda M. Cytokines and chemokines in irritant contact dermatitis. *Mediators Inflamm*. 2013; 2013:916497

REVIEWERS' COMMENTS:

Reviewer #2 (Remarks to the Author):

I am satisfied with answers/comments provided by authors.

Reviewer #3 (Remarks to the Author):

I think the authors have done a good job of addressing reviewer's comments. I have no further critiques or comments to add.